## RESEARCH ARTICLE

# UBQLN2 is necessary for UBE3A-mediated proteasomal degradation of the domesticated retroelement PEG10

Julia E. Roberts[1,*], Phuoc T. Huynh[1,2,*], Luis O. Carale[1] and Alexandra M. Whiteley[1,‡]

## ABSTRACT

Ubiquilins are a family of extrinsic ubiquitin receptors that are thought to facilitate protein degradation by shuttling proteins to the proteasome. However, the defining characteristics of ubiquilin clients, and the steps of ubiquilin-mediated degradation, have been elusive. Previously, we showed that ubiquilin 2 (UBQLN2) regulates the proteasomal degradation of PEG10, a unique virus-like protein that comes in two forms: a gag protein, which is not regulated by UBQLN2, and a gag-pol protein, which is dependent on UBQLN2. Here, we refine the model of ubiquilin activity through further investigation of the UBQLN2-mediated degradation of PEG10. Gag-pol and gag proteins undergo distinct degradation processes; both forms bind to UBQLN2 independently of their ubiquitylation status, but only gag-pol protein is degraded in a UBQLN2-, ubiquitin- and proteasome-dependent fashion. Cellular gag-pol is ubiquitylated, and mutation of key lysine residues in the pol region rendered gag-pol insensitive to UBQLN2. Degradation of gag-pol was also dependent on the E3 ubiquitin ligase UBE3A, which requires UBQLN2 to regulate gag-pol levels. Together, these data clarify our understanding of UBQLN2-mediated degradation and highlight the importance of UBE3A in regulating PEG10.

KEY WORDS: Ubiquilin, UBQLN2, Proteasome, UBE3A, E6AP, PEG10

## INTRODUCTION

Protein degradation is essential to cellular and organismal health, and the majority of proteins within the cell are degraded by the ubiquitin–proteasome system (UPS). In this pathway, E1, E2 and E3 enzymes mark proteins for degradation through the covalent attachment of ubiquitin chains at one or more lysine residues (Hershko et al., 1983; Hershko and Ciechanover, 1998; Nandi et al., 2006). Intrinsic ubiquitin receptors located in the proteasome regulatory cap recognize ubiquitylated proteins and facilitate docking and subsequent threading of proteins into the core for proteolytic cleavage (Finley, 2009; Lander et al., 2012).

While many proteins are degraded efficiently through the UPS, a subset of proteasome clients rely on the assistance of extrinsic

ubiquitin receptors, or 'shuttle factors'. These receptors bind to the proteasome through a ubiquitin-like (UBL) domain (Elsasser et al., 2002; Chen et al., 2016; Buel et al., 2023) and bind to ubiquitin moieties through a ubiquitin-associated (UBA) domain (Funakoshi et al., 2002; Kang et al., 2006; Harman and Monteiro, 2019). Ubiquilins (UBQLNs) are the largest mammalian family of extrinsic ubiquitin receptors, and additionally contain either one or two intervening STI1 domains that facilitate client binding and protein–protein interactions (Hjerpe et al., 2016; Suzuki and Kawahara, 2016; Kurlawala et al., 2017; Zheng et al., 2020; Fry et al., 2021; Onwunma et al., 2025 preprint). There are five ubiquilins in humans: ubiquilins 1, 2, 3, 4 and L, which all share the same general protein domain structure, but diverge in tissue expression (Marin, 2014; Lin et al., 2022). UBQLN2 has also gained recent attention as gene variants are associated with a familial form of amyotrophic lateral sclerosis with frontotemporal dementia (Deng et al., 2011; Wu et al., 2020; Lin et al., 2022).

An unresolved question within the ubiquilin field has been the identity and molecular property of their clients. In yeast, loss of the *UBQLN* ortholog *Dsk2* results in a proteome-wide accumulation of K48-linked ubiquitylated proteins (Funakoshi et al., 2002), which supports the notion that Dsk2 is a general facilitator of ubiquitylated protein degradation. However, mammalian ubiquilins may play a more complex role. Mammalian ubiquilins bind with nearly equal affinity to a variety of ubiquitin linkages (Zhang et al., 2008), and recent work has demonstrated ubiquilin-dependent proteasomal degradation of non-ubiquitylated proteins (Makaros et al., 2023), suggesting a more complicated model of their function.

In addition to this unresolved question of client selection, the field lacks a clear chronological understanding of how ubiquilins mediate client degradation. Although ubiquilins were traditionally thought to act after the substrate has already undergone ubiquitylation, more recent evidence has challenged this notion and points towards a putative model in which the UBQLN–client interaction recruits enzymes to initiate ubiquitylation of the client protein (Itakura et al., 2016; Onwunma et al., 2025 preprint; Scheutzow et al., 2024 preprint), thereby facilitating either degradation or protection of proteins from breakdown.

Among the multiple E3 ligases linked to ubiquilins is the HECT E3 ligase UBE3A (also known as E6AP) (Kleijnen et al., 2000; Martínez-Noël et al., 2018). UBE3A is well known for its oncogenic role in the proteasomal degradation of P53 upon binding to the human papillomavirus E6 protein (Yamamoto et al., 1997); however, variants of the *UBE3A* gene are also associated with the neurodevelopmental disease Angelman's syndrome (Kishino et al., 1997). UBE3A regulates proteins involved in AMPA receptor recycling (Drebushenko et al., 2025), including the retrotransposon-derived gag-like protein Arc (Greer et al., 2010). Structural work has shown the N-terminal AZUL domain of UBE3A binds to the UBA of UBQLN1 and UBQLN2 (Buel et al., 2023), but how

[1]Department of Biochemistry, University of Colorado Boulder, Boulder, CO 80309, USA. [2]Department of Molecular, Cellular, and Developmental Biology, University of Colorado Boulder, Boulder, CO 80309, USA.
*These authors contributed equally to this work

‡Author for correspondence (Alexandra.whiteley@colorado.edu)

J.E.R., 0000-0002-9222-6132; P.T.H., 0000-0001-7364-9191; L.O.C., 0009-0004-2776-5940; A.M.W., 0000-0002-4144-7605

UBQLN2 and UBE3A cooperate to modulate proteostasis remains unknown.

In this study, we used the human protein PEG10 as a reporter to map out the process of UBQLN2-mediated degradation. Paternally expressed gene 10 (*PEG10*) is a unique retrotransposon-derived gene that possesses many structural features of retrotransposons, but has lost the enzymatic domains necessary for genetic duplication (Brandt et al., 2005). However, it has retained a −1 programmed ribosomal frameshift, which allows for the formation of two functionally unique protein products from a single mRNA (Brandt et al., 2005; Manktelow, 2005; Clark et al., 2007; Lux et al., 2010), which we refer to as gag and gag-pol. The *PEG10* mRNA contains a pseudoknot just before the gag stop codon (Manktelow, 2005) that stalls translation while the ribosome unwinds RNA. During the stall, the ribosome rests on a 'slippery sequence' of nucleotides that results in inefficient −1 ribosome frameshifting during the pause (Namy et al., 2006). If frameshifting does not occur, the ribosome unwinds the pseudoknot and translates gag protein. If frameshifting occurs, the gag stop codon is out of frame, and the ribosome will synthesize the gag-pol fusion protein (Clark et al., 2007; Lux et al., 2010). Whereas gag-pol is a client of UBQLN2, gag is not (Whiteley et al., 2021; Black et al., 2023; Mohan et al., 2025). Further, UBE3A regulates degradation of gag-pol, but not gag (Pandya et al., 2021). Together, these provide a unique opportunity to explore ubiquilin client identity and the details of ubiquilin-mediated degradation.

## RESULTS

### PEG10 gag-pol degradation is dependent on the ubiquitin–proteasome system and autophagy

While many reported ubiquilin clients are destined for degradation through the UPS (Hjerpe et al., 2016; Itakura et al., 2016), some clients are degraded in a ubiquitin-independent pathway (Makaros et al., 2023), and others are degraded through autophagy (Rothenberg et al., 2010; Yun Lee et al., 2013; Wu et al., 2020; Idera et al., 2023). To determine the pathway of degradation for PEG10, endogenous gag and gag-pol protein abundance was probed in wild-type (WT) HEK293 cells upon proteasome and E1 inhibition by bortezomib and TAK243, respectively. As expected, poly-ubiquitylated proteins increased in total cell lysate upon bortezomib treatment due to the inhibition of proteasomal degradation (Fig. 1A). Conversely, poly-ubiquitylated proteins were decreased, while mono-ubiquitin accumulated, upon TAK243 treatment (Fig. 1A). Treatment of cells with both TAK243 and bortezomib prevented the accumulation of poly-ubiquitylated protein (Fig. 1A). Gag protein levels were not significantly impacted by either drug (Fig. 1A,B) but gag-pol accumulated significantly upon either proteasome or E1 inhibition (Fig. 1A,B), indicating that gag-pol abundance is uniquely regulated by ubiquitylation and proteasomal degradation.

Accumulation of gag-pol upon proteasome and E1 inhibition suggested that the protein may be ubiquitylated as a prerequisite for degradation by the proteasome. To monitor ubiquitylation of

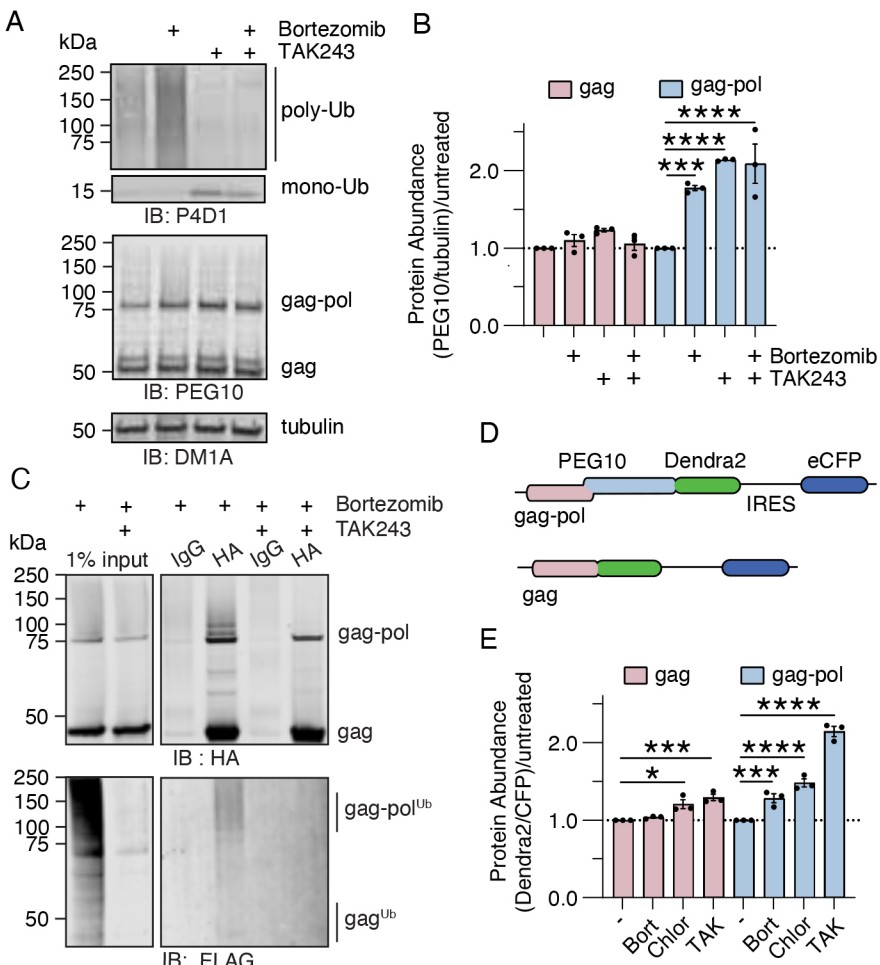

**Fig. 1. PEG10 gag-pol degradation is dependent on the ubiquitin–proteasome system and autophagy.** (A) Western blot of WT HEK293 whole-cell lysate. Cells were treated with drug (either 50 nM bortezomib or 1 μM TAK243) for 16 h prior to harvest. Shown is one of three representative experiments. (B) Quantification of the data shown in A. Samples were normalized to the abundance of gag or gag-pol protein in untreated cells for each individual experiment. Statistics were determined by two-way-ANOVA with Šídák's multiple comparisons, *n*=3. Differences in gag treatments were not statistically significant. (C) Representative western blot analysis following IP using HA-tag antibody or IgG control from whole-cell lysate of cells stably expressing HA-gag-pol and transiently transfected with 3×FLAG-Ubiquitin. Cells were treated with drug for 16 h prior to harvest and IP. Shown is one of two representative experiments. (D) Schematic of the Dendra2 reporter. Gag is shown in pink, and pol is shown in blue. Dendra2 is fused at the C terminus to gag-pol and will only be visible when the ribosome has successfully frameshifted. eCFP, enhanced CFP; IRES, internal ribosome entry site. (E) Client accumulation assay performed using gag and gag-pol constructs, with and without drug treatment. Cells were treated with 50 nM bortezomib, 100 μM chloroquine or 1 μM TAK243 for 16 h prior to flow cytometry. Like B, samples were normalized to the abundance of each protein in untreated (−) wells for each individual experiment. Statistics were determined by two-way-ANOVA with Šídák's multiple comparisons, *n*=3. For B,E, data are shown as mean ±s.e.m. *P<0.05, ***P<0.001, ****P<0.0001. IB, immunoblot.

PEG10 more directly, a 3×FLAG-Ubiquitin (3×FLAG-Ub) plasmid was transfected into a cell line stably expressing HA-tagged PEG10. Cells were treated with bortezomib to enhance PEG10 yield for native immunoprecipitation (IP) in the presence or absence of TAK243. When probed against HA, a ladder of high molecular weight gag-pol was visible with bortezomib treatment, with clear bands at ~90 and 100 kDa, which disappeared upon co-treatment with TAK243 (Fig. 1C, top). A smear of 3×FLAG-Ub at the exact same molecular weight was visible in HA-precipitated samples (Fig. 1C, bottom), which disappeared upon TAK243 treatment. A limitation is that native IP may co-precipitate PEG10 binding partners that are also ubiquitylated; however, the close agreement between HA- and FLAG-tagged probes strongly suggest that it is HA-tagged PEG10 that is ubiquitylated. Probing of PEG10 IPs with antibodies generated against acetylated lysine (Fig. S1A) or polyADP-ribose (Fig. S1B) did not show any enrichment of signal compared to input, suggesting that the major modification of lysine residues of PEG10 appears to be ubiquitin. In contrast, minimal ubiquitin signal was visible at or near the molecular weight for gag (Fig. 1C).

Similar results were found using a fluorescent reporter of PEG10 accumulation. In brief, fusion constructs were generated between proteins of interest and a Dendra2 fluorescent protein, followed by an IRES-CFP cassette (Fig. 1D). Cells were transfected with either gag-Dendra2 or gag-pol-Dendra2 constructs, and flow cytometry was performed 48 h later. The Dendra2/CFP ratio provides transfection-controlled protein abundance values with single-cell granularity (Fig. S1C). As gag abundance was not impacted by proteasome or E1 inhibition in our previous experiment, an inhibitor of autophagy, chloroquine, was added. In this assay, gag protein accumulated slightly upon inhibition of autophagy and ubiquitylation (Fig. 1E; Fig. S1C). In contrast, PEG10 gag-pol accumulated upon inhibition of the proteasome, autophagy and ubiquitylation (Fig. 1E; Fig. S1C).

## Proteasomal regulation of PEG10 gag-pol is ubiquilin dependent

To determine how ubiquilins contribute to these protein degradation pathways, the same assays were performed in a triple-knockout ubiquilin HEK293 cell line lacking ubiquilins 1, 2 and 4 (TKO; Itakura et al., 2016). First, endogenous levels of PEG10 were quantified by western blotting in TKO cells after proteasome or E1 inhibition (Fig. 2A). In contrast to WT HEK cells, gag-pol was not regulated by the proteasome but was impacted by E1 inhibition in TKO cells (Fig. 2B), implying abundance was regulated by ubiquitylation but not by proteasomal degradation. TKO cells also showed a significant increase in gag protein upon treatment with TAK243, but not with bortezomib alone (Fig. 2B).

To compare more directly with WT cells, TKO cells were treated with bortezomib, chloroquine or TAK243, and probed on the same blot (Fig. 2C). In WT cells, gag-pol accumulated dramatically upon bortezomib treatment and TAK243 treatment; however, in this assay, chloroquine had no measurable effect on endogenously expressed gag-pol (Fig. 2D). In contrast, gag-pol levels were comparatively high in resting TKO cells and were unaffected by any drug treatment (Fig. 2D). Gag levels were only mildly affected by drug treatment and were not changed by ubiquilin expression (Fig. S2A). In the flow-based client accumulation assay, gag-pol abundance did not change upon proteasomal inhibition in TKO cells (Fig. 2E; Fig. S2B). When measured by flow cytometry, chloroquine treatment of TKO cells still resulted in gag-pol accumulation (Fig. 2E), leading us to conclude that autophagic degradation of gag-pol is ubiquilin independent and varies depending on the assay used.

To summarize our findings, paired data from Figs 1E and 2E of transfected WT and TKO cells were transformed to obtain TKO/WT values, where ubiquilin clients that accumulate in TKO cells have values >1, whereas values ≤1 represent non-clients. Gag showed no regulation by ubiquilins regardless of drug treatment (Fig. 2F, left), as evidenced by the values of ~1 in all culture conditions. Without drug treatment, gag-pol levels were higher in TKO cells, which was mitigated by inhibition of the proteasome or ubiquitylation (Fig. 2F, right). Treatment with chloroquine did not diminish the contribution of ubiquilins to gag-pol abundance, indicating that autophagy and lysosomal degradation of PEG10 are not significantly impacted by the presence or absence of ubiquilins.

To assess the PEG10 half-life, a cycloheximide chase was performed on WT and TKO cells in the presence of bortezomib, chloroquine or TAK243 for 20 h. After 20 h of cycloheximide, both WT and TKO cells had lost approximately 30% of gag-pol signal (Fig. S2C,D) and 70% of gag (Fig. S2C,E). In this time frame, bortezomib and chloroquine had no significant effect on the half-life of gag-pol for either cell line. In addition, there was no apparent difference in the half-life of gag-pol between WT and TKO cells (Fig. S2D). We posit that, given the long half-life of gag-pol, cycloheximide chases may not capture the contribution of UBQLN2 to degradation; however, further work is warranted to investigate this possibility.

It is possible that deletion of ubiquilins could have unexpected effects on global translation. Therefore, rates of PEG10 protein synthesis were also examined. Cells were starved of methionine and incubated with the methionine analog, azido-homo alanine (AHA), which was labeled after 4 h of protein synthesis with biotin-alkyne. There were no overall differences in the rate of global protein synthesis between WT and TKO cells as measured in whole-cell lysate (Fig. S3A). PEG10 protein was then immunoprecipitated, and the strength of biotin-AHA and PEG10 bands were detected by western blotting (Fig. S3B). In both cell lines, there was a low level of AHA incorporation into gag-pol protein, which was visibly decreased in TKO cells (Fig. S3B). There were also no differences in the synthesis of gag (Fig. S3B). From this, we conclude that TKO cells do not synthesize more gag-pol than WT cells, and that the high level of gag-pol protein is more likely due to a failed degradation process than changes to synthesis.

## Lysine residues of the pol region of PEG10 are essential for UBQLN2-dependent regulation

PEG10 contains 15 lysine residues in its gag region, one lysine in the pseudoknot and four (393, 567, 586, 590) in its pol region (Fig. 3A). Previous work shows many of these residues are ubiquitylated in human cell lines (Hornbeck et al., 2015; Abed et al., 2019), but it is not known how individual lysines influence protein stability. By mutating lysine residues to arginine (K>R), the contribution of specific residues to UPS-mediated degradation can be directly interrogated. Because of the unique ubiquilin dependence of gag-pol protein, lysines in the pol region of PEG10 were subjected to mutation first. WT and TKO cells were transfected with K>R mutant reporter constructs, and protein abundance was plotted for each as TKO/WT. When all four pol lysines were mutated to arginine, gag-pol lost all dependence on ubiquilins for its degradation (Fig. 3B,D). However, no single K>R mutations of pol lysines had any significant effect on ubiquilin-dependent accumulation (Fig. 3B). When two K>R mutations were combined, they had variable effects, suggesting a dominant role of K567 (Fig. 3C,D). When three residues were mutated K>R, dependence on ubiquilins was also completely lost (Fig. 3C,D), implying a secondary role of K393. Consistent with a role for ubiquilins in facilitating

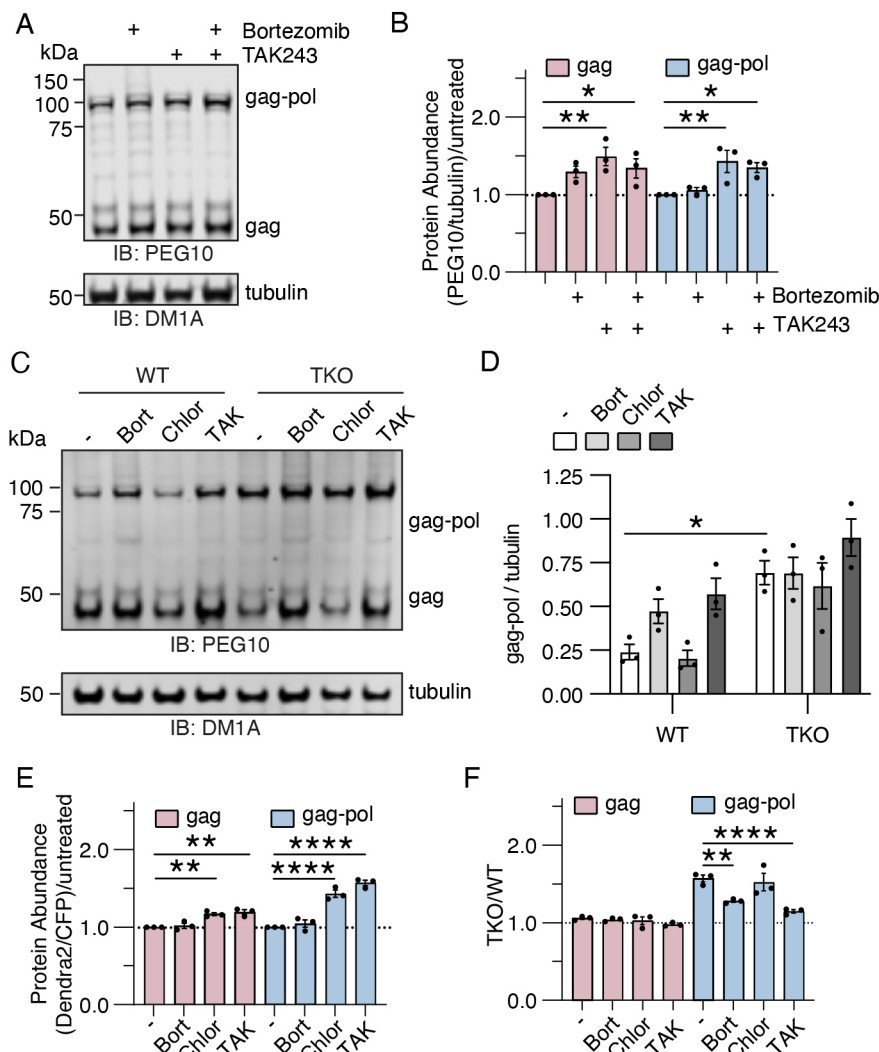

**Fig. 2. Proteasomal, but not autophagic, degradation of gag-pol is ubiquilin dependent.**
(A) Western blot of TKO HEK293 whole cell lysate. Cells were treated with drug for 16 h prior to harvesting as in Fig. 1. Shown is one of three representative experiments. (B) Quantification of the data shown in A. Samples were normalized to the abundance of each protein in untreated cells for each individual experiment. Statistics were determined by two-way ANOVA with Šídák's multiple comparisons, n=3. (C) WT and TKO cells were treated with the stated drugs as in Fig. 1 for 20 h and then lysed for western blotting of PEG10. Shown is one of three representative blots. (D) Quantification of the results from C, n=3. Statistics were determined by two-way ANOVA with Šídák's multiple comparisons. (E) Plotted data for flow-based client accumulation assay performed in TKO cells using gag and gag-pol with and without (−) drug treatment for 16 h prior to flow cytometry. Like B, samples were normalized to the abundance of each protein in untreated cells for each individual experiment. Statistics were determined by two-way ANOVA with Šídák's multiple comparisons, n=3. (F) TKO/WT values were generated from paired experimental samples from Figs 1E and 2E by dividing the raw TKO Dendra/CFP values by WT Dendra/CFP values for each independent experiment. A TKO/WT ratio of 1.0 indicates that PEG10 abundance is unchanged between the two cell lines. Statistics were determined by two-way ANOVA with Šídák's multiple comparisons, n=3. For B,D–F, data are shown as mean±s.e.m. *P<0.05, **P<0.01, ****P<0.0001. IB, immunoblot.

degradation by the ubiquitin–proteasome pathway, PEG10 gag-pol$^{K>R}$ did not meaningfully accumulate upon proteasome inhibition, but did accumulate upon autophagy or E1 inhibition (Fig. 3E). Together, they suggest that mutation of pol region lysine residues significantly impacts ubiquilin-dependent degradation via the UPS.

To determine the contribution of lysines in the gag region to PEG10 degradation, all lysine residues within gag were mutated to arginine. As expected, gag$^{K>R}$ was not a client of ubiquilins (Fig. S4); however, gag$^{K>R}$-pol was also not a client of ubiquilins (Fig. S4), indicating that at least one lysine residue in the gag region is also required for ubiquilin-mediated degradation.

### Binding of UBQLN2 and PEG10 is multi-domain and dependent on gag

The classical model of ubiquilin-mediated protein degradation involves an interaction between the ubiquilin and the tetra-ubiquitylated client protein to facilitate proteasomal delivery. IP of UBQLN2 in the presence of the crosslinking reagent DSP resulted in the co-precipitation of both PEG10 gag and gag-pol (Fig. 4A), consistent with previous reports (Mohan et al., 2025) and highlight the finding that interaction with UBQLN2 is not sufficient to render gag a client. PSMD4 (also known as Rpn10), a subunit of the proteasome and a known UBQLN2 interactor (Ko et al., 2004; Hamazaki et al., 2015), was also visible (Fig. 4A). To exclude an

indirect contribution of other ubiquilins to a UBQLN2–PEG10 interaction, a TKO cell line with doxycycline-inducible expression of MYC-UBQLN2 was used for IP. As before, IP of MYC-UBQLN2 co-precipitated both gag and gag-pol protein (Fig. S5A, left), and IP of PEG10 resulted in co-precipitation of MYC-UBQLN2 (Fig. S5A, right). IP of endogenous PEG10 also resulted in a reciprocal co-precipitation of UBQLN2 (Fig. 4B). Notably, although UBQLN2 and UBQLN1 are known to interact (Ford and Monteiro, 2006; Lee and Brown, 2012), UBQLN1 was not enriched by UBQLN2 or PEG10 IP (Fig. 4A,B), demonstrating the selectivity of the interaction between UBQLN2 and PEG10.

To test whether ubiquitylation was important for this interaction, MYC-UBQLN2-expressing cells were treated with TAK243 for an endogenous PEG10 IP. MYC-UBQLN2 was present regardless of TAK243 treatment (Fig. 4C), indicating that ubiquitylation is not necessary for the interaction between UBQLN2 and PEG10. Cells were then transfected with HA-tagged K>R mutants and 3×FLAG-Ub, followed by an IP of HA-tagged PEG10. These constructs still co-precipitated UBQLN2, indicating that lysine-mediated ubiquitylation of PEG10 is not necessary for interaction with UBQLN2 (Fig. 4D). When the pattern of ubiquitin staining was examined, a slight but reproducible decrease in ubiquitin banding was observed with gag-pol$^{K>R}$ but not with gag$^{K>R}$-pol (Fig. 4D), although it remains possible that co-precipitating proteins contribute

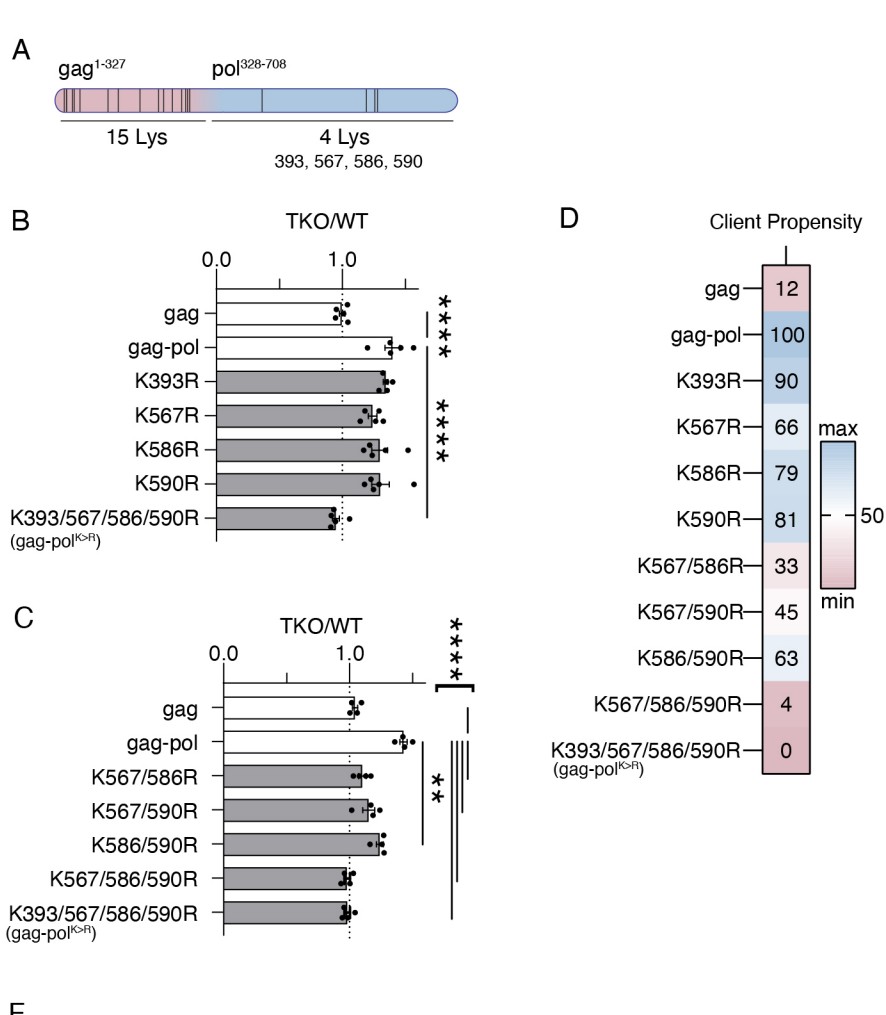

**Fig. 3. PEG10 pol lysine residues are necessary for ubiquilin-mediated degradation.** (A) Schematic showing the approximate location of lysine residues along PEG10 gag-pol. (B,C) Plotted cumulative data for client accumulation assay of single K>R mutations (B) and combinatorial K>R mutations (C). All experimental data are shown, but statistics were calculated by one-way ANOVA with Dunnett's multiple comparisons using gag-pol as a control and only paired samples within an experiment. (D) Client propensity heatmap on the right was calculated by averaging all TKO/WT values for the experiments in B and C, normalized to the minimum and maximum of the dataset. (E) K393/567/586/590R (gag-pol$^{K>R}$) was subjected to the client accumulation assay. Cells were treated with or without (−) the drugs shown for 16 h prior to flow cytometry. Samples were normalized to the abundance of each protein in untreated cells for each individual experiment. Statistics were determined by two-way ANOVA with Šídák's multiple comparisons, $n=3$. For B,C,E, data are shown as mean±s.e.m. *$P<0.05$, **$P<0.01$, ***$P<0.001$, ****$P<0.0001$.

to the observed ubiquitin signal. Whereas ubiquitylation is necessary for gag-pol degradation, it is dispensable for interaction with UBQLN2. Consistent with this finding, we observed that TKO cells expressing FLAG-tagged UBQLN2$^{ΔUBA}$ or UBQLN2$^{ΔUBL}$ (Gerson et al., 2020) were both capable of co-precipitating endogenous PEG10 with UBQLN2 (Fig. S5B).

Beyond UBA–ubiquitin interactions, Ubiquilin–client binding is thought to be governed by STI1–client protein interactions. Surprisingly, deletion of both STI1 domains also had negligible effects on binding to PEG10 (Fig. 4E). Because of the selective nature of the UBQLN2–PEG10 interaction, we surmised that the PXX domain, which is unique to UBQLN2, may play a larger role. FLAG-UBQLN2$^{ΔPXX}$ was expressed in TKO cells and immunoprecipitated, then probed for PEG10. There was a reproducible decrease in the amount of co-precipitated gag-pol protein, indicating that the PXX domain partially contributes to PEG10 binding (Fig. 4F). The amyotrophic lateral sclerosis-associated *UBQLN2* variants P497H,

which has a minor effect on PEG10 abundance, and P506T, which has a more significant effect on PEG10 abundance (Black et al., 2023), did not completely abolish the interaction with PEG10 (Fig. S5C), although it remains possible there are minor changes to affinity that are not readily apparent in these tests.

## UBE3A, an E3 ligase and binding partner of UBQLN2, alters gag-pol ubiquitylation and abundance

While ubiquitylation of PEG10 is dispensable for its interaction with UBQLN2, it is still necessary for ubiquilin-dependent, proteasomal degradation (Figs 1 and 3). Furthermore, the interaction between gag-pol and UBQLN2 is not sufficient for degradation to occur, as gag-pol$^{K>R}$ interacts with UBQLN2, but is not degraded by the proteasome. One study of ubiquilin-mediated degradation determined that binding of Omp25 by UBQLN1 resulted in recruitment of an unknown E3 ligase to facilitate ubiquitylation and subsequent degradation of the protein (Itakura et al., 2016). Recent work has

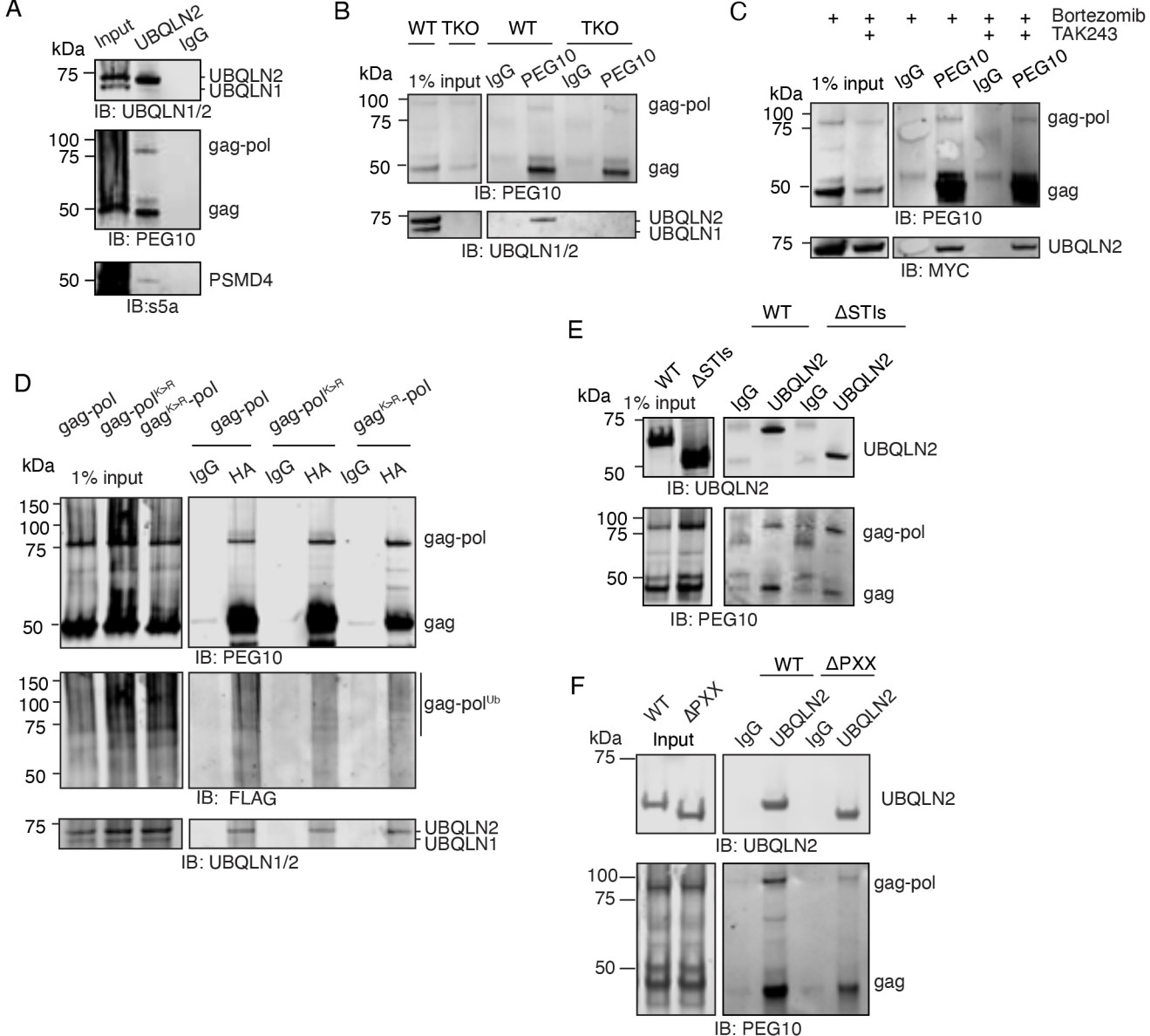

**Fig. 4. UBQLN2 interacts with PEG10 in a ubiquitin-independent manner.** (A) IP of endogenous UBQLN2 using UBQLN2-specific antibody or IgG control from WT cells followed by western blot analysis shows co-precipitation of PEG10 gag and gag-pol, as well as the proteasomal subunit PSMD4. The UBQLN1/2 antibody used for western blotting recognizes both ubiquilins, but the antibody used for IP only binds to UBQLN2. PEG10 was detected with a polyclonal antibody. Shown is one of two representative experiments. (B) WT and TKO HEK293 cells were subjected to an IP of endogenous PEG10 using PEG10 polyclonal antibody or IgG control followed by western blot analysis. Western blot was probed for PEG10 with polyclonal antibody and UBQLN1/2 using an antibody that recognizes both ubiquilins. Shown is one of two representative experiments. (C) MYC-UBQLN2-expressing cells were drug treated for 16 h, and lysate was subjected to an IP using a PEG10 or IgG control antibody followed by western blot analysis. PEG10 was detected with a polyclonal antibody and UBQLN2 was detected by anti-MYC-tag antibody. Shown is one of two representative experiments using bortezomib; $n$=1 for TAK243. (D) WT HEK293 cells were transiently transfected with HA-tagged PEG10 constructs and a plasmid expressing 3×FLAG-Ub. An IP was performed with either HA-tag antibody or IgG control and samples were visualized by western blotting. PEG10 was probed with HA antibody, ubiquitin was probed with FLAG antibody, and UBQLN2 was visualized with an antibody that recognizes both UBQLN1 and UBQLN2. Shown is one of two representative experiments. (E) Both STI1 domains were removed from FLAG-UBQLN2 (ΔSTIs) and protein was again immunoprecipitated with UBQLN2 antibody following transfection of TKO HEK293 cells. PEG10 was detected with a polyclonal antibody. Shown is one of two representative experiments. (F) The UBQLN2 PXX domain was deleted (ΔPXX) and protein was immunoprecipitated with UBQLN2 antibody from transfected TKO HEK293 cells. PEG10 was detected with a polyclonal antibody. Shown is one of two representative experiments. IB, immunoblot.

also shown that UBQLN2 mediates the protection of ATP5G1 from degradation through facilitated interaction with the E3 ligase SCF[bxo7] (Scheutzow et al., 2024 preprint). Considering these findings, our results led us to examine how UBE3A, an E3 ligase previously linked to both UBQLN2 and PEG10, may work to facilitate PEG10 degradation via UBQLN2.

IP of UBQLN2 co-precipitated UBE3A, PEG10 and the proteasome subunit PSMD4 (Fig. 5A). To examine the contribution of the AZUL domain to this interaction, we transfected cells with FLAG-tagged UBE3A WT (UBE3A[FL]) or ΔAZUL (UBE3A[ΔAZUL]) and performed the same IP. Both constructs co-precipitated with UBQLN2 and PEG10 (Fig. 5B), although UBE3A[ΔAZUL] may have a

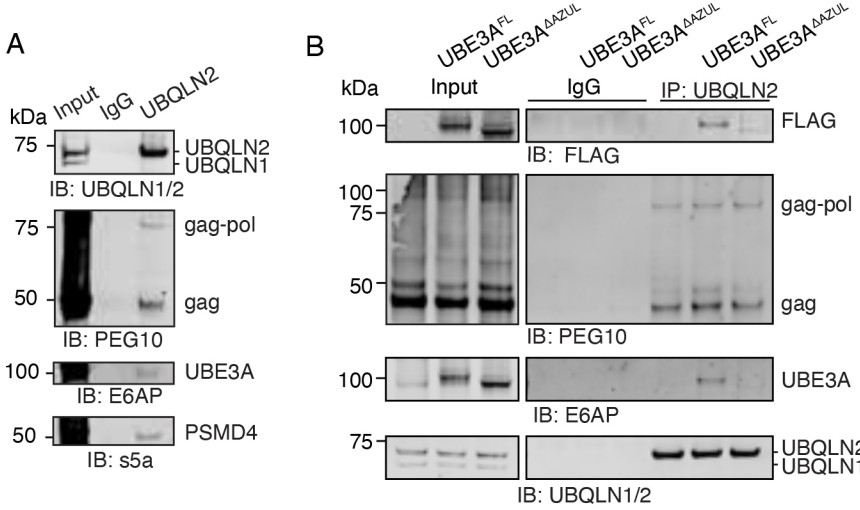

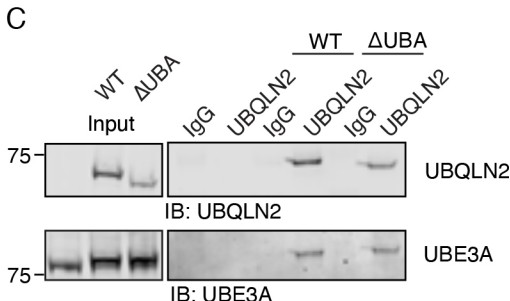

**Fig. 5. The UBE3A AZUL domain is important for interaction with UBQLN2.** (A) WT HEK293 cell lysate was subjected to an IP with anti-UBQLN2 antibody or IgG control and detected by western blotting. UBQLN1/2 were detected using an antibody that recognizes both ubiquilins but was precipitated with an antibody that only recognizes UBQLN2. $n$=2. (B) WT HEK293 were transfected with FLAG-tagged UBE3A constructs. After 48 h, cells were lysed and UBQLN2 was immunoprecipitated by anti-UBQLN2 antibody or IgG control, followed by western blotting. UBE3A was detected with FLAG antibody and PEG10 was detected with a polyclonal antibody. $n$=3. (C) UBE3A binds to UBQLN2 in a UBA-independent manner. TKO HEK293 cells were co-transfected with the FLAG-tagged UBE3A and FLAG-UBQLN2 constructs shown. UBQLN2 was immunoprecipitated using a UBQLN2 antibody and probed against UBE3A with a UBE3A antibody. $n$=2. IB, immunoblot.

partial defect in UBQLN2 binding. Despite the slight decrease in UBE3A$^{\Delta AZUL}$, PEG10 proteins co-precipitated at equal amounts with UBQLN2 (Fig. 5B), suggesting that the interaction between UBQLN2 and PEG10 is independent of UBE3A. To determine the contribution of the UBA domain of UBQLN2 to UBE3A binding, an IP was performed for either WT or UBQLN2$^{\Delta UBA}$ and probed for UBE3A. UBE3A immunoprecipitated with both UBQLN2 constructs (Fig. 5C), indicating that the UBA domain is not necessary for co-precipitation with UBE3A; however, UBA-adjacent sequences may make up for the loss of the UBA domain (Buel et al., 2023). In conclusion, UBE3A associates with UBQLN2, and, in this system, its interaction is partially dependent on the presence of the AZUL domain.

*UBE3A* depletion experiments confirmed a role for the E3 ligase in regulating gag-pol abundance and revealed a dependence on ubiquilins for this control. WT and TKO HEK293 cells were transfected with non-targeting control or *UBE3A*-targeting siRNAs followed by western blotting for endogenous PEG10 (Fig. 6A–D; Fig. S6A–D). In WT cells, knockdown of *UBE3A* did not influence gag protein levels (Fig. 6C, left) but increased gag-pol abundance significantly (Fig. 6D, left). In contrast, *UBE3A* knockdown had almost no effect on gag-pol levels in TKO cells (Fig. 6D; Fig. S6D), suggesting that UBE3A does not regulate PEG10 in the absence of ubiquilins.

To determine whether UBE3A was sufficient to regulate gag-pol abundance, a rescue experiment was performed. CRISPR/Cas9 knockouts of *UBE3A* expression were generated in ubiquilin-TKO HEK293 cells and confirmed by DNA sequencing (Fig. S6E) and western blotting (Fig. 6E). TKO cells and TKO/UBE3A knockout cells had comparable levels of PEG10 protein at the steady state (Fig. 6E), which would be consistent with UBQLN2 being required for UBE3A-mediated control of PEG10. For the rescue experiment,

WT FLAG-UBE3A or FLAG-UBE3A$^{\Delta AZUL}$ was transfected alone or with FLAG-UBQLN2, then harvested for western blotting (Fig. 6F). In cells with intact UBE3A but no UBQLN2, transfection with additional WT UBE3A mildly decreased gag-pol levels (Fig. 6G). Transfection of these cells with UBE3A lacking the AZUL domain resulted in higher gag-pol levels (Fig. 6G), indicating that this domain may be important for gag-pol control. In contrast, TKO/UBE3A knockout cells transfected with UBE3A constructs alone had a minor decrease in gag-pol (Fig. 6G). In both cell lines, co-transfection of UBQLN2 strongly suppressed gag-pol abundance, irrespective of the presence of the AZUL domain of UBE3A (Fig. 6G). In all conditions, gag levels were unaffected (Fig. S6F). These results suggest that the presence of UBQLN2 is essential for UBE3A to control gag-pol abundance; together with previous data, these suggest a pathway whereby UBQLN2 and UBE3A work together to facilitate ubiquitin–proteasome-mediated degradation of gag-pol protein.

## DISCUSSION

Here, we have explored UBQLN2-dependent proteasomal degradation of the retrovirus-like protein, PEG10. Two alternative PEG10 proteins are formed resulting from the infrequent utilization of a −1 ribosome frameshift: a short form, referred to as gag, which is not a client of UBQLN2, and a long form, gag-pol, which requires UBQLN2 for its degradation (Whiteley et al., 2021; Black et al., 2023; Mohan et al., 2025).

The first notable finding of this study is that gag and gag-pol undergo distinct degradation processes. While the abundance of gag was not affected by inhibition of the proteasome, it was mildly increased upon treatment with chloroquine, and shutdown of global ubiquitylation. Conversely, inhibition of the UPS led to an

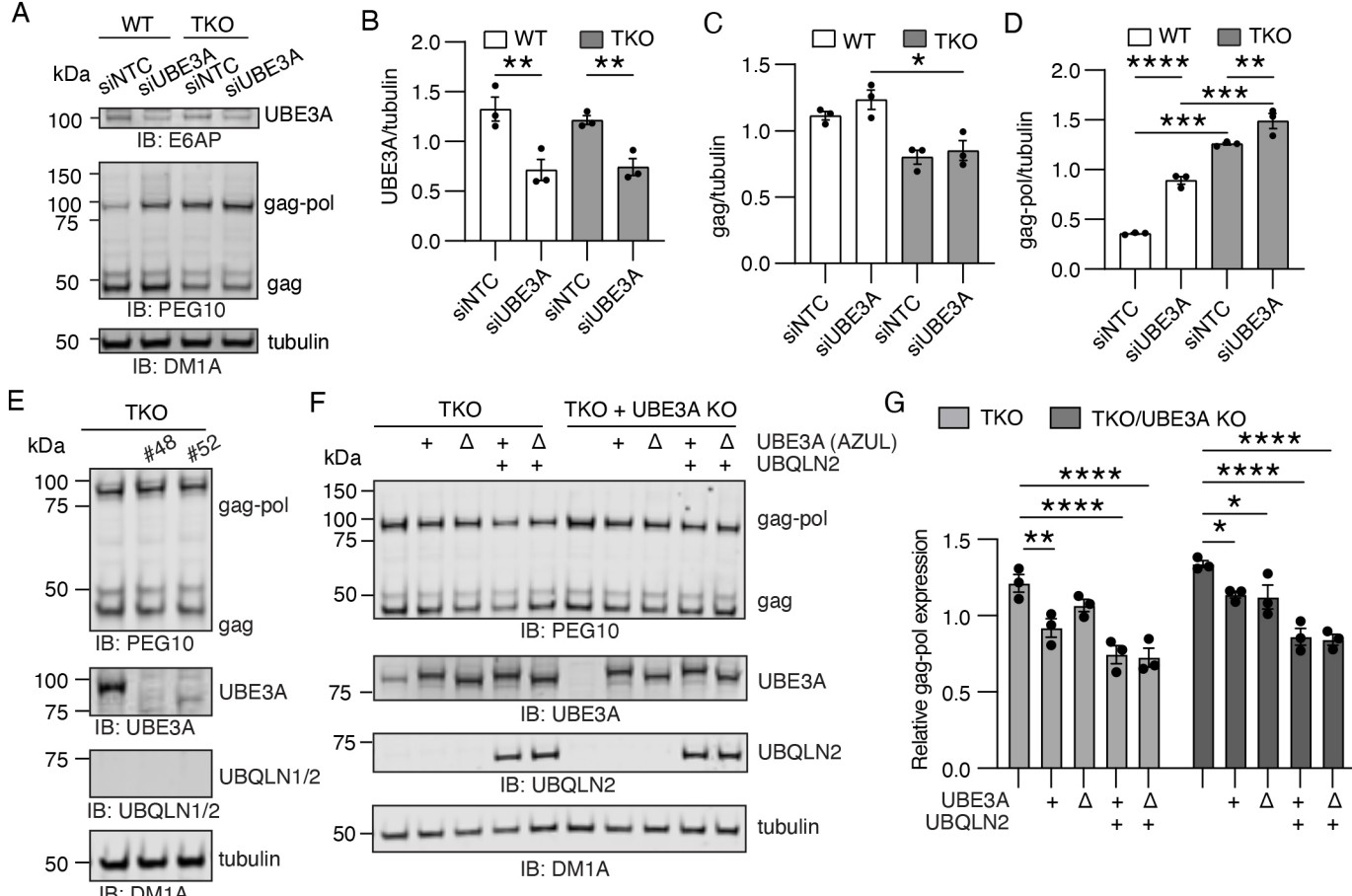

**Fig. 6. UBE3A and UBQLN2 are both required to control levels of gag-pol protein.** (A) Representative western blot of *UBE3A* knockdown in WT and TKO HEK293 cells. Cells were transfected with 100 pmol siRNA against a non-targeting construct (NTC) or *UBE3A* and harvested for western blotting 72 h later. PEG10 and UBE3A were visualized with polyclonal antibodies. Shown is one of three representative experiments. (B) Quantification of UBE3A protein abundance from the experiment shown in A. (C) Quantification of gag protein abundance from the experiment shown in A. (D) Gag-pol protein abundance from the experiment shown in A. For B–D, results for each band were normalized to a blot average of that band for each independent experiment, and statistics were determined by two-way ANOVA with Šídák's multiple comparisons. Shown is mean±s.e.m., *n*=3. (E) Western blot of CRISPR/Cas9 *UBE3A* knockout cell lines. TKO HEK293 cells were used as a genetic background, so no ubiquilin expression is evident. (F) Rescue experiment of TKO/UBE3A knockout cell line with transfection of UBE3A and UBQLN2 constructs. Delta symbol signifies transfection of UBE3A$^{\Delta AZUL}$. Cells were transfected for 48 h and lysed for western blotting using polyclonal PEG10 and UBE3A antibodies, and a UBQLN2 antibody that recognizes both UBQLN1 and UBQLN2. Shown is one of three representative experiments using clone #48. (G) Quantification of gag-pol protein levels from the experiment shown in F. Delta symbol signifies transfection of UBE3A$^{\Delta AZUL}$. Gag-pol levels were quantified, normalized to tubulin for each well, and then normalized to a blot average. Shown is mean±s.e.m. for three independent experiments. Statistics were determined by two-way ANOVA with Šídák's multiple comparisons. *$P<0.05$, **$P<0.01$, ***$P<0.001$, ****$P<0.0001$. IB, immunoblot.

accumulation of gag-pol. This difference was further reflected in their apparent ubiquitylation status: whereas gag-pol was substantially ubiquitylated, gag was not. The proteasome-dependent regulation of gag-pol was additionally dependent on the presence of ubiquilins, and likely UBQLN2 in particular. Previous work has shown that deletion of only *UBQLN2* is sufficient to modulate gag-pol levels (Black et al., 2023; Mohan et al., 2025), and, in this study, UBQLN2, but not UBQLN1, co-precipitated with PEG10.

UBQLN2 co-precipitated with both gag and gag-pol proteins, but only gag-pol degradation is dependent on UBQLN2. Others have also observed UBQLN2 interaction with gag protein even though gag does not rely on ubiquilins for its regulation (Mohan et al., 2022, 2025). This suggests that interacting with a ubiquilin is not sufficient to result in proteasomal degradation of that protein. Given the importance of particular lysine residues in the pol region for effective proteasomal degradation, we propose that the additional requirement for PEG10 is ubiquitylation.

Although ubiquilins are capable of facilitating ubiquitin-independent proteasomal degradation (Makaros et al., 2023), the ubiquilin-mediated degradation of gag-pol hinged on functional ubiquitylation. Immunoprecipitated PEG10 showed a smear of ubiquitylation at, and above, the molecular weight of gag-pol, whereas gag had minimal ubiquitin staining. In particular, when HA-tagged PEG10 was immunoprecipitated, we observed bands at ∼90 and 100 kDa, which could represent intermediates in the process of multi-ubiquitylation. Consistent with these findings, three lysine residues (567, 586, 590) within the pol region were necessary for the degradation of the gag-pol protein, suggesting that their ubiquitylation contributes to proteasomal degradation. Mutation of the four lysine residues of pol led to a decrease in apparent ubiquitin staining, which suggests that these lysines account for some of the visible ubiquitin pattern of gag-pol protein. Ubiquitylation of the pol region was not sufficient for proteasomal degradation, as a gag$^{K>R}$-pol mutant was also defective in degradation. We propose

that ubiquitylation of both gag and pol regions is necessary but not sufficient for degradation by the proteasome, which could explain why gag is not a client despite interacting with UBQLN2 (Mohan et al., 2025). However, there are three limitations of these findings. First, it is possible that ubiquitin staining comes from co-precipitating proteins of the same molecular weight as PEG10. Second, it remains possible that K>R mutations could alter the structure of PEG10 in a way that does not influence UBQLN2 binding but still eliminates UBQLN2-mediated degradation. In addition, we cannot rule out the possibility that other unexplored post-translational modifications also contribute to ubiquilin-dependent degradation.

Probing of ubiquilin–client interactions has demonstrated a contribution of STI1 domains (Fry et al., 2021; Onwunma et al., 2025 preprint; Acharya et al., 2025) as well as UBA–ubiquitin interactions (Kurlawala et al., 2017). For PEG10, we found that the UBA–ubiquitin interaction and STI1 domains both appeared dispensable for the interaction between UBQLN2 and PEG10; in contrast, we found a contribution of the PXX domain. Although unexpected, these findings could explain why UBQLN2, but not UBQLN1, co-precipitated with PEG10, and why modulation of UBQLN1 levels fails to influence PEG10 abundance (Black et al., 2023). While UBQLN1 and UBQLN2 share their overall domain structure, only UBQLN2 has the unique PXX domain (Marin, 2014), which could confer specific regulation of PEG10. Mutation of the PXX domain leads to increased PEG10 levels (Black et al., 2023) and may slightly, but not completely, abrogate PEG10 binding; however, future work is needed to explore this hypothesis further. Future work should also focus on the cooperative role of RTL8 in the UBQLN2-dependent regulation of PEG10 (Mohan et al., 2022, 2025), and how this fits with UBE3A-dependent ubiquitylation.

Previous studies have shown PEG10 ubiquitylation is regulated by the HECT E3 ligase UBE3A, and that UBE3A binds to UBQLN2 (Kleijnen et al., 2000; Pandya et al., 2021; Buel et al., 2023). *UBE3A* knockdown in WT HEK293 cells correlated with an increase in gag-pol. In contrast, *UBE3A* knockdown had a minimal effect on gag-pol abundance in ubiquilin-deficient cells, which suggests that UBQLN2 is necessary for the UBE3A-mediated proteasomal degradation of gag-pol protein. This could explain why previous attempts to observe PEG10 ubiquitylation by UBE3A *in vitro* were unsuccessful (Pandya et al., 2021), although more work is necessary to confirm that UBQLN2 is sufficient to promote direct ubiquitylation of PEG10 by UBE3A.

Our model supports a multistep pathway of ubiquilin-mediated proteasomal degradation. We propose that UBQLN2 binds to a non-ubiquitylated PEG10, and UBE3A. UBE3A then ubiquitylates gag-pol, which is necessary for its proteasomal degradation. This stands in contrast to previous models in which UBA–ubiquitin interactions dominate ubiquilin–client binding, and emphasizes the requirement for involvement of the E3 ligase in defining the degradation pathway of the ubiquilin client. Our current understanding and model show a need for the interactions of multiple domains and unique residues of UBQLN2, PEG10 and UBE3A that are necessary for association and subsequent protein degradation. Additionally, this body of work shows the client specificity of UBQLN2, and highlights the possibility of other specific ubiquilin, client and E3 ligase partners.

## MATERIALS AND METHODS
### Cloning
All *Homo sapiens* PEG10 [gag (AA 1-325), gag-pol (AA 1-708)] and UBE3A isoform 2 [FL (AA 1-853) ΔAZUL (AA 60-853); Kleijnen et al., 2000] plasmids were cloned into pCDNA3.1 or pDendra2 backbone designed with a CMV promoter, using GeneArt Gibson Assembly (Invitrogen). For

PEG10 lysine-to-arginine mutants of the pol region, site directed mutagenesis was used. PEG10 gag$^{K>R}$ was purchased from Twist Biosciences before PCR and Gibson assembly to generate gag$^{K>R}$-pol constructs. All HA-tagged constructs had an N-terminal 2xHA-tag. All FLAG-tag constructs made in this study had an N-terminal 3×FLAG-tag. All Dendra2-tagged constructs had a C-terminal Dendra2 tag. pCMV4-FLAG-UBQLN2 [WT (AA 1-624), ΔUBL (AA 1-624 missing AA 33-107), ΔUBA (AA 1-580)] and ΔPXX FLAG-UBQLN2 (missing AA 491-526) were a gift from Dr Lisa Sharkey (Kleijnen et al., 2000; Mohan et al., 2022). ΔSTIs FLAG-UBQLN2 (missing AA 178-248 and AA 379-462) was generated by Gibson cloning. The 3×FLAG-Ubiquitin plasmid generated in this paper was subcloned from an HA-Ubiquitin plasmid (Addgene plasmid #18712). WT *UBE3A* was amplified directly from HEK293 cDNA, then Gibson primers were designed to ensure inclusion or deletion of the AZUL domain.

Chemically competent DH5α *Escherichia coli* (Invitrogen) were transformed with plasmid and plated on either 50 μg/ml kanamycin (Teknova, K2150) or 100 μg/ml carbenicillin (Gold Biotechnology, C-103-5) in LB agar (Teknova, L9115) for 16-18 h. Singles colonies were selected and sequenced by either Sanger sequencing (Azenta) or whole-plasmid sequencing (Plasmidsaurus). Sequence-confirmed plasmids were midi-prepped (Zymo Research, D4201) and stored at approximately 1 μg/μl of purified DNA in DEPC-treated water in preparation for mammalian cell transfection.

### Cell culture and cell lines
WT and *UBQLN1*, *UBQLN2* and *UBQLN4* TKO HEK293 cells, and Flp-In T-REx 293 TKO cells expressing a doxycycline-inducible MYC-UBQLN2 (WT, P497H, P506T) were a gift from Dr Ramanujan Hegde (Medical Research Council Laboratory of Molecular Biology, Cambridge, UK). Flp-In HEK293 cells (Invitrogen) expressing HA-gag-pol were generated in-house using manufacturer's recommendations. Cells were maintained following standard cell culture protocols, at 37°C with 5% CO$_2$, using Dulbecco's modified Eagle's medium (Gibco, 11965092) supplemented with 1% penicillin/streptomycin (Invitrogen), 1% L-glutamine (R&D Systems, Inc.), and 10% fetal bovine serum (FBS) (Millipore Sigma). The concentration of contaminating doxycycline in standard cell culture FBS was sufficient to induce approximately endogenous levels of expression of MYC-UBQLN2 in Flp-In cells (Black et al., 2023).

For drug treatments, cells were treated for 16-20 h using final concentrations of 50 nM bortezomib (EMD Millipore, 5043140001), 100 μM chloroquine (diphosphate; VWR, 22113-5G), or 1 μM TAK243 (Selleck Chemicals, S8341).

### Transfection
HEK293 cells were grown to 80% confluency in cell culture dishes with a variety of well sizes depending on the end-use of the cells. Using manufacturer recommendations, Lipofectamine 2000 (Invitrogen, 11668027) was used at a DNA (μg) to Lipofectamine 2000 (μl) ratio of 1:2.5, diluted in 1× Opti-MEM (Gibco, 31985062).

### Western blotting
Cells were harvested 48 h after transfection by gentle pipetting and spun once in PBS to wash pellets. Cell pellets were lysed in 8 M urea lysis buffer (8 M urea, 75 mM NaCl, 50 mM Tris-HCl pH 8.5, 1× cOmplete Mini EDTA-free protease inhibitor cocktail tablet). Lysate was centrifuged for 10 min at 21,300 *g* and supernatant was collected as whole-cell lysate.

Total protein in lysate was quantified by BCA protein assay (Pierce BCA Protein Assay kit, 23227) and then samples were added to Laemmli sample buffer with β-mercaptoethanol (Sigma-Aldrich) for SDS-PAGE gel electrophoresis. Proteins were separated on 4-12% NuPage Bis-Tris pre-cast gels (Invitrogen), and transferred to either nitrocellulose (Cytiva, 10600009) or Immobilon PVDF (specifically for detection of ubiquitin) membranes at 15 V for 60 min (Invitrogen). For detecting endogenous proteins, 10 μg of total protein was used per biological sample. For detecting transfected proteins, 5 μg of total protein was used.

Membranes were blocked using LI-COR Intercept TBS or PBS blocking buffer for 30 min, washed three times for 5 min each wash using 1× Tris-buffered saline with 0.1% Tween-20 (TBS-T), and detected using the LI-

COR IR secondary antibody system. Primary antibodies for PEG10 (Proteintech 14412-1-AP; 1:1000), HA-tag (either Proteintech 66006-2-Ig or Cell Signaling Technology C29F4; 1:5000), UBQLN2/1 (5F5 clone, Novus Biologicals; 1:1000), UBQLN2 (6H9 clone, Novus Biologicals; 1:1000), M2 FLAG-tag (Sigma-Aldrich F3165; 1:5000), UBE3A (Proteintech 10344-1-AP; 1:1000), PSMD4 (Cell Signaling Technology 12441S; 1:1000), MYC tag (Cell Signaling Technology 2276S; 1:2000), acetylated lysine (CST Ac-K-100 6952S; 1:1000), poly/mono-ADP Ribose (D9P7Z clone, Cell Signaling Technology; 1:1000) and tubulin (DM1A clone, Novus Biologicals; 1:10,000) were left on overnight at 4°C, while secondary antibodies (IR680 and IR800 conjugates, LI-COR) were incubated for 1 h at room temperature at 1:10,000 concentration. Protein bands were visualized using the LI-COR Odyssey CLx and image analysis performed using LI-COR ImageStudio Software. Full uncropped blots are shown in Fig. S7.

### Flow cytometry – client accumulation assay
Cells were transfected in 48-well plates and harvested 48 h after transfection by pipette mixing using 200 µl Flow Buffer (D-PBS, 2% FBS, 0.1% sodium azide). Samples were transferred into U-bottom 96-well plates and analyzed using a BD FACSCelesta fitted with a high-throughput sampler with the following settings: sample flow rate (1.0 µl/s), sample volume (100 µl), mixing volume (100 µl), mixing speed (200 µl/s), number of mixes (2), wash volume (200 µl). All experimental data were analyzed using FlowJo.

### Gating strategy
Transfected cells were initially gated in FSC-A vs SSC-A with the polygon gating tool to identify 'cells'. Within this 'cells' population, CFP-positive cells were gated in 405 nm versus SSC-A. A novel parameter to derive Dendra2 Green/CFP was created by dividing 488 references by 405 references, with a logarithmic scale, min=0.0001 and max=10. This custom parameter was utilized only in CFP-positive cells, and the geometric mean was exported and used to generate all graphs.

### Metabolic labeling
HEK293 cells were plated in 10-cm dishes and starved of methionine using methionine depletion media made of DMEM high glucose, −Glu, −Cys, −Met (Gibco, 21013024), L-cysteine (Gibco), 1% L-Glu (Gibco), 10% dialyzed FBS (Gibco, A3382001), for 45 min followed by treatment with 2 µM AHA (Life Technologies) for 4 h. After incubation, cells were harvested and lysed using a gentle IP lysis buffer (50 mM Tris-HCl pH 7.4, 150 mM NaCl, 1% Triton X-100, 1× cOmplete EDTA-free protease inhibition) and lysate was cleared by centrifugation at 16,000 $g$ for 5 min at 4°C. To label the total AHA sample, PEG4 biotin-alkyne was conjugated using manufacturer's recommendations and proteins were methanol: chloroform precipitated for western blot.

IP of PEG10 was as described below, with the additional step of performing the biotin-alkyne click-reaction directly on the resin before elution. Following binding of protein to beads overnight, the Thermo Fisher Click-It protein labeling kit was used at ½ volumes to add biotin-alkyne to nascent AHA-containing proteins. After labeling, tubes were washed with IP wash buffer three times, followed by protein elution using Laemmli buffer and heating at 95°C for 10 min.

### IP and co-IP
IPs and co-IPs were performed under non-denaturing conditions. Antibodies against PEG10 (Proteintech 14412-1-AP), HA-tag (Cell Signaling Technology C29F4), UBQLN2 (6H9 clone, Novus Biologicals), M2 FLAG-tag (Sigma-Aldrich F3165), or IgG control (BioLegend 910801) were conjugated to protein A or G resin beads (50% slurry) at a ratio of 1:8, using 40 µl of resin and 5 µg of antibody per IP condition. Antibodies were incubated with the beads for 30-60 min at room temperature, before buffer exchange into 0.2 M sodium borate (pH 9.0). Antibodies were crosslinked to the beads using 20 mM DMP diluted in 0.2 M sodium borate (pH 9.0) for 30 min at room temperature with rocking. The volume of DMP solution used was ten times the resin bed volume (40 µl 50% slurry=400 µl DMP solution). The crosslinking reaction was terminated using 0.2 M ethanolamine (pH 8.0), which was left overnight at 4°C. The next day, the resin was washed

three times in 1× D-PBS before adding lysate. Resin beads were spun down at 3000 $g$, for 3 min.

For most co-IPs, cells were plated in 10 cm dishes and treated with bortezomib for 16 h to induce accumulation of protein products. An exception is that the endogenous UBQLN2 co-IPs shown in Figs 4 and 6 were not subjected to proteasome inhibitor treatment prior to lysis. Cells were lysed in 500 µl Triton lysis buffer (200 mM NaCl, 10 mM HEPES, 10 mM EGTA, 10 mM EDTA, 1% Triton X-100, 1× cOmplete EDTA-free protease inhibitor), and centrifuged at 16,000 $g$ for 5 min at 4°C. Supernatant was collected and equally split between IgG and antibody-conjugated resin and incubated at 4°C overnight with an end-over-end rotator. The next day, the resin was washed three times for 2 min each wash, using IP Wash Buffer (1× D-PBS, 0.1% Triton X-100, 1× cOmplete EDTA-free protease inhibitor). Protein elution was achieved by the addition of IP Elution Buffer (2× Laemmli in 1× PBS), and boiling the resin for 15 min at 95°C. Centrifugation (3000 $g$ for 3 min) was used to pellet the resin, and eluate was collected before being visualized by western blotting.

### DSP crosslinking for co-IPs
For co-IP experiments, cells were treated with DSP (Lomant's Reagent, Thermo Fisher Scientific) in one of two manners (Akaki et al., 2022). Cells were collected in tubes at 300 $g$ for 3 min and washed in 1× D-PBS. Right before use, 50 mM DSP was dissolved in DMSO, and then further diluted in 1× D-PBS for a final concentration of 1 mM. Crosslinking was performed for 30 min at room temperature with rocking. DSP-mediated crosslinking was stopped using DSP stop buffer (20 mM Tris-HCl pH 7.5 in PBS) for 15 min at room temperature with rocking. Cells were washed three times with 1× D-PBS, before lysis as previously described for IP. Alternatively, cells were washed and treated with DSP directly in 10-cm dishes before harvest and lysis prior to co-IP.

### CRISPR/Cas9 gene editing
Ubiquilin TKO HEK293 cells were used for CRISPR/Cas9 gene editing. Guide RNA directed against *UBE3A* (CRISPR673067_SGM) was purchased from Life Technologies. TKO cells were plated in a 6-well plate and transfected with 5 µg recombinant TrueCut Cas9 protein v2 and 37.5 pmol of selected TrueGuide Synthetic gRNA using CRISPRMax transfection reagent according to the manufacturer's recommendations (Life Technologies). One week after transfection, cells were diluted to 0.8 cells/100 µl and plated at single-cell dilution in 96-well plates. Media was changed as needed over the course of approximately 1 month until cells could be transferred to 12-well plates for outgrowth and western blotting. Once confluent, cells were harvested for western blotting against endogenous UBE3A protein. Clones with no apparent UBE3A band compared to control HEK293 cells were grown further for gDNA isolation using the QIAGEN DNeasy kit. PCR primers flanking the gRNA site were selected and used to PCR amplify gDNA isolated from each clone. Then, purified PCR products (one clear band from clone #48 and two bands close in size from clone #52) were sent for long-read amplicon sequencing (Plasmidsaurus). Following confirmation of *UBE3A* gene editing, cells were expanded for freezing and further experimentation.

### Statistical analysis
Statistical analyses were performed utilizing the integrated analysis suite in GraphPad Prism. Analyses used include standard one-way ANOVA followed by Dunnett's multiple comparisons test, and two-way ANOVA followed by Šídák's multiple comparisons test. Statistical tests used are listed in all figure legends, error bars represent mean±s.e.m., and $P$-values are represented as *$P<0.05$, **$P<0.01$, ***$P<0.001$ and ****$P<0.0001$.

### Acknowledgements
We thank Autumn M. Matthews for her excellent work in making and validating the stably expressed PEG10 HA-gag-pol Flp-In HEK293 cells. We also acknowledge the Department of Biochemistry at CU Boulder: Cell Culture Facility [RRID: SCR_018988 (CCF)], the Flow Cytometry Shared Facility (RRID: SCR_019309) and the Shared Instruments Pool (RRID: SCR_018986).

### Competing interests
A.M.W. is a co-founder of Endios Bio and is an author on patents related to PEG10 inhibition. All other authors declare no competing or financial interests.

## Author contributions

Conceptualization: A.M.W.; Data curation: J.E.R., P.T.H.; Formal analysis: J.E.R., P.T.H., A.M.W.; Funding acquisition: A.M.W.; Investigation: J.E.R., P.T.H.; Methodology: J.E.R., P.T.H., L.O.C., A.M.W.; Project administration: A.M.W.; Supervision: A.M.W.; Validation: J.E.R., L.O.C., P.T.H.; Visualization: J.E.R., L.O.C., P.T.H.; Writing – original draft: J.E.R., A.M.W.; Writing – review & editing: J.E.R., P.T.H., L.O.C., A.M.W.

## Funding

This work was supported by the National Institute of Neurological Disorders and Stroke (R01NS131660 to A.M.W.) and the National Institute of General Medical Sciences (T32 GM142607 to J.E.R. and P.T.H.). Open Access funding provided by the University of Colorado. Deposited in PMC for immediate release.

## Data and resource availability

All relevant data and details of resources can be found within the article and its supplementary information.

## Peer review history

The peer review history is available online at https://journals.biologists.com/jcs/lookup/doi/10.1242/jcs.264105.reviewer-comments.pdf

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
