## [Peer Review File · Journal of Cell Science]

UBQLN2 is necessary for UBE3A-mediated proteasomal degradation of the domesticated retroelement PEG10

Julia E. Roberts, Phuoc T. Huynh, Luis O. Carale and Alexandra M. Whiteley
DOI: 10.1242/jcs.264105

Editor: Pedro Carvalho

Review timeline

Original submission:	23 April 2025
Editorial decision:	1 June 2025
First revision received:	14 October 2025
Editorial decision:	24 October 2025
Second revision received:	5 November 2025
Accepted:	8 November 2025

Original submission

First decision letter

MS ID#: jcs.264105

MS TITLE: UBQLN2 facilitates degradation of the retrotransposon protein PEG10 via UBE3A activity

AUTHORS: Julia Roberts; Phuoc Huynh; Alexandra Whiteley

ARTICLE TYPE: Research Article

Dear Dr Whiteley,

We have now reached a decision on the above manuscript.

To see the reviewers' reports and a copy of this decision letter, please go to:

Reviewer 1

SUMMARY OF THE ADVANCE MADE IN THIS PAPER AND ITS POTENTIAL SIGNIFICANCE TO THE FIELD

The manuscript by Roberts J and colleagues reported a role for the ubiquitin binding protein UBQLN2 in the degradation of retrotransposon protein PEG10 (gag-pol). In a previous screen, the senior author identified UBQLN2 as a modulator of PEG10 stability, but the underlying mechanism was unclear. In the current study, they further characterized the PEG10 degradation mechanism. They showed that treating cells with proteasome- and ubiquitin E1-inhibitors causes this protein to accumulate in WT cells, but not in UBQLN triple KO cells. Co-IP demonstrated an interaction between UBQLN2 and gag-pol, but surprisingly, the gag protein, another translation product of PEG10, bound to UBQLN2 much better than gag-pol, although gag is not targeted for degradation by UBQLN2. Further, the interaction between UBQLN2 and gag-pol is not dependent on substrate ubiquitination. Through a mutagenesis study, they identified a few lysine residues that are required for gag-pol stability regulation, but no evidence suggests that these mutations affect substrate ubiquitination. Their knockdown experiment further implicated E6AP, a HECT domain-containing E3 ubiquitin ligase in gag-pol regulation, but knockdown of E6AP does not seem to abolish substrate ubiquitination. Although the study provides some evidence that further elucidates the mechanism by which gag-pol is degraded in cells, the reported findings do not seem to present a clear picture

on the precise role played by UBQLN2 in gag-pol degradation. There are many gaps here and there in the study. The following questions should be addressed to make the study more complete.

SUGGESTIONS TO AUTHORS

Main points:

1. The title claims Ubiquitin 2 regulates gag-pol degradation via E6AP, but most of the study was done with a triple knockout cell line that lacks three ubiquitin members. How can they conclude that it is Ubiquitin 2 that regulates gag-pol. The title also implies that E6AP (UBE3A) acts downstream of Ubiquitin 2, but there is no evidence for this.
2. Figure 1A should provide the drug concentration information. The accumulation of gag-pol after 16h of proteasome inhibition is not very impressive. How do they know that this is due to protein stabilization as opposed to regulations at other levels (e.g. transcription, RNA stability, translation)? The presentation of Fig 1B (and other quantification data) is not appropriate. The y-axis does not start from 0, which is misleading.
3. Page 4, line 21, the authors claimed that only high molecular weight 120-250kDa signal was increased in cells treated with a proteasome inhibitor. However, from the data in Fig. 1C, it seems that in addition to the high molecular weight smear, there are two clear bands at ~90 and 100 kDa positions in bortezomib treated cells. There is no information on whether these bands are present in untreated cells. Fig. S1A and B quantified 4 different conditions, but the representative gel in C only showed 2 conditions. It would be nice to include the gel data for the other two conditions.
4. Figure 1C, was this IP experiment done under denaturing conditions? The experimental procedure does not include any information.
5. When addressing the role of ubiquitin in gag-pol degradation (Fig. 2), they used an unconventional experimental design that paired two independent experiments together (Fig. 2D). A more straightforward way to look at this is to directly measure the half life of gag-pol in WT and Ubiquitin knockout cells using cycloheximide chase.
6. Fig 2E shows that gag binds ubiquitin 2 much better than gag-pol. They should discuss why this substrate is not degraded.
7. Figure 3, they should explore how these lysine mutants affect the ubiquitination status of gag-pol. This is critical for the interpretation of their data because it is strange that mutating lysines in either the gag or pol region can "stabilize" this protein. In general, if lysines play redundant roles in protein turnover, one has to mutate all of them to see a phenotype. It is possible that some lysine mutations may affect protein via a ubiquitin independent mechanism.
8. Page 8, lines 15-38, the lengthy introduction about UBE3A should be placed in the introduction section.
9. Please provide representative blots for Fig. 5E. Does UBE3A knockdown affect the low molecular weight ubiquitinated gag-pol species? Why did knockdown of an ubiquitin ligase cause substrate to accumulate in ubiquitinated form. This result raises the question of the role of UBE3A in gag-pol regulation and suggests that the model in their graphic abstract is oversimplified.
10. The co-IP experiment does not prove a direct interaction between UBE3A and gag-pol. There is no evidence to suggest that Ubiquitin, UBE3A and substrate can form a ternary complex.
11. The paper should be shortened and revised to improve the clarity and readability. The authors should pay close attention to their conclusions to ensure they are logical and without over-interpretation. For example, on page 11, lines 21-23. These sentences are confusing. I don't understand what the authors meant by "there were comparatively minimal changes to TKO cells". Didn't they just show that TKO cells had more gag-pol in Fig. 5A? How can they conclude here that UBE3A functions downstream of UBQLN2? The next sentence is even more confusing. They claimed that "UBQLN2 is necessary for UBE3A-mediated ubiquitination of PEG10". Where is the evidence? The data on Fig. 5E clearly showed the opposite (the knockdown of UBE3A INCREASED gag-pol ubiquitination). The data is against their model in the graphic abstract. The discussion on the liquid-liquid phase separation of ubiquitin is completely irrelevant to the study.

Reviewer 2

SUMMARY OF THE ADVANCE MADE IN THIS PAPER AND ITS POTENTIAL SIGNIFICANCE TO THE FIELD
 Ubiquitins are a family of UBL/UBA containing proteins that are known to shuttle ubiquitinated proteins to the proteasome for degradation. Building off previous work showing that the domesticated retrotransposon PEG10 is a UBQLN2 client, Roberts et al provide a detailed analysis

of the different steps for UBQLN2 mediated degradation of PEG10. Key results include demonstrating that Ubiquilins mediate proteasomal degradation of PEG10, showing that PEG10-UBQLN2 interaction is ubiquitin-independent, identification of key lysine residues in PEG10, and the discovery that the UBE3A-UBQLN2 interaction facilitates PEG10 ubiquitination. Overall, I find the results to be relatively rigorous and, after revision, appropriate for publication. Given the complexity of the system and the mixed results (such as proteasomal vs. autophagy pathways, multiple E3 ligases, multiple ubiquitination events required), the paper could benefit from rewriting to better acknowledge nuance in the system, and some clarifying experiments.

SUGGESTIONS TO AUTHORS

Major Concerns

- * While it is clear that some portion of gag-pol is degraded via the proteasome and Ubiquilins mediate this function, I am not convinced that the proteasome is the major degradation pathway. Instead, the results in figure 1E indicate that the majority of gag-pol degradation may be occurring via autophagy. Many substrates are known to show plasticity in which degradation pathway is utilized, with a shift occurring upon inhibition of one pathway. The following changes are recommended.
 - o Repeat the experiments in 1A-C with chloroquine.
 - o Identify the ubiquitin linkage on PEG10 after proteasomal inhibition. Is it K48 linked? This is optional, but it could provide some clarity about proteasomal degradation vs. autophagy.
 - o Language will need to change to indicate that while degradation can occur via the UPS, this is not the exclusive (or even dominant) pathway.
- * An alternative model is that the gag to gag-pol ratio is changing due to differences in translation rather than differences in protein stability. It is difficult to differentiate between these models because figure 1 only looks at steady state levels. The following changes are recommended.
 - o Given the extensive comparisons between the stability of gag vs. gag-pol, the article would benefit from enhanced introduction of translation of PEG10, including what determines the ratio of translation gag vs. gag-pol protein.
 - o Unless the gag to gag-pol translation ratio is known to be stable across many stress conditions, Figure 1 should be repeated as a pulse-chase or cycloheximide chase to look at protein stability.
 - o I know that the authors have used the Dendra/CFP assay previously to study PEG10 stability. However, the figure and description could benefit from some clarification. For example, it's not readily clear how gag levels were measured in figure 1E. I assume it is a separate construct that only encodes Gag-Dendra and lacks the Pol region.

Minor Concerns

- * Please label western blots to indicate which antibody is used (HA, Flag, gag-pol, UBQLN2, etc.). I know this is in the figure legend, but it would help if it is directly labeled in the figure.
- * Statistics for many of the bar graphs are far away from the bar and it takes some effort to match the statistics with the appropriate bars (for example, Figure 2B). If possible, please adjust.
- * Figure legend 1A states that this is a representative experiment twice.
- * In figure 1B, are the results with gag statistically significant? Perhaps explicitly state that only significant values are labeled.
- * Please add a supplemental figure showing representative flow cytometry data for figure 1E and 2C. It is important to see the Dendra2/CFP ratio for the entire population of cells rather than just the average.
- * Please be more explicit for how 2D was prepared. For example, it appears that it is just the Dendra/CFP ratio for TKO/WT. Or is it the (Dendra/CFP)/untreated for TKO/WT?
- * Figure 6A, please indicate number of replicates.
- * The abstract states that ubiquitination is dependent of Ube3a, but the results show that other E3 ligases are involved. The abstract should be rewritten accordingly.
- * The authors frequently state that gag is not a ubiquilin client, but the data clearly shows gag interacting with UBQLN2. The authors should be explicit about what it means to be a client (shuttled for degradation vs. binding).

Reviewer 3

SUMMARY OF THE ADVANCE MADE IN THIS PAPER AND ITS POTENTIAL SIGNIFICANCE TO THE FIELD

This study investigates the degradation of the gag-pol fusion protein encoded by PEG10. Earlier work had shown that PEG10 degradation occurs via UBQLN2 and the proteasome. In the current work, the authors build on this in order to gain additional insights into how UBQLN2 selects clients and how it targets them for degradation. The field is a bit muddled on this issue, with models ranging from UBQLNs selecting clients after they have been ubiquitinated, to ones in which UBQLNs select clients and facilitate their ubiquitination. The most recent, and arguably most rigorous and compelling, studies indicate that UBQLNs using their STI domains to recognise exposed hydrophobicity, then use the UBA domain to recruit E3 ligase(s), and finally use the UBL domain to target the substrate to the proteasome. This model was elucidated in PMID 27345149, with further structural analysis of the STI domains in PMID 33620121 & 39026758, and characterisation of the client range in PMID 28933694.

In the context of current knowledge, the main advance of this work is to characterise another substrate (PEG10) and implicate UBE3A as one of the Ubiquilin-dependent ligases involved in substrate degradation. The characterisation of PEG10 degradation as being UBQLN, ubiquitin, and proteasome dependent is generally clear, but is a modest advance beyond earlier work. The latter part on UBE3A is a bit underdeveloped. If this can be improved, and a few additional controls provided as noted below, the paper should be suitable for publication.

SUGGESTIONS TO AUTHORS

Specific points:

- 1) (minor) Fig. 1 and 2 are baseline characterisation of PEG10 degradation, and are generally clear and convincing. However, it would be useful to have had direct comparisons of WT versus TKO cells analysed on the same gel in at least some of these initial experiments so a reader can directly appreciate the magnitude of the difference in steady state levels relative to suitable loading controls. This can be seen in Fig. 5A, but as a reader, I would like to have seen such a direct comparison early in the paper.
- 2) (major) Figure 4 would benefit from analysis of a UBQLN2 variant in which one or more of the STI domains are deleted to directly test whether these regions are needed as hypothesized. This is an important conclusion of the study - that client recognition is not via ubiquitin or the UBA domain - so actually identifying the region of UBQLN2 that recognises clients seems important and central to the study.
- 3) (major) The siRNA result for UBE3A is not entirely convincing without a rescue. It would be better if UBE3A were knocked out with CRISPR and rescued with either wild type or Δ AZUL versions of UBE3A. This will not only establish specificity of the effect, but would also test whether the AZUL domain is functionally important. As the UBE3A part of the study is the main new advance, it seems reasonable to establish this point more thoroughly and to provide at least some basic molecular insights into which part of UBE3A interacts with which part of UBQLN2.
- 4) (major) Related to point 3, the prediction is that Ubiquilin's UBA domain is responsible for interacting with UBE3A to facilitate substrate ubiquitination. This seems worthwhile to test by analysing the interaction of WT versus Δ UBA versions of UBQLN2 in co-IP experiments.

First revision

Author response to reviewers' comments

Reviewer 1: SUMMARY OF THE ADVANCE MADE IN THIS PAPER AND ITS POTENTIAL SIGNIFICANCE TO THE FIELD

The manuscript by Roberts J and colleagues reported a role for the ubiquitin binding protein UBQLN2 in the degradation of retrotransposon protein PEG10 (gag-pol). In a previous screen, the

senior author identified UBQLN2 as a modulator of PEG10 stability, but the underlying mechanism was unclear. In the current study, they further characterized the PEG10 degradation mechanism. They showed that treating cells with proteasome- and ubiquitin E1-inhibitors causes this protein to accumulate in WT cells, but not in UBQLN triple KO cells. Co-IP demonstrated an interaction between UBQLN2 and gag-pol, but surprisingly, the gag protein, another translation product of PEG10, bound to UBQLN2 much better than gag-pol, although gag is not targeted for degradation by UBQLN2. Further, the interaction between UBQLN2 and gag-pol is not dependent on substrate ubiquitination. Through a mutagenesis study, they identified a few lysine residues that are required for gag-pol stability regulation, but no evidence suggests that these mutations affect substrate ubiquitination.

Their knockdown experiment further implicated E6AP, a HECT domain-containing E3 ubiquitin ligase in gag-pol regulation, but knockdown of E6AP does not seem to abolish substrate ubiquitination. Although the study provides some evidence that further elucidates the mechanism by which gag-pol is degraded in cells, the reported findings do not seem to present a clear picture on the precise role played by UBQLN2 in gag-pol degradation. There are many gaps here and there in the study. The following questions should be addressed to make the study more complete.

We thank the reviewer for this thorough summary and thoughtful critique, and have tried to address their concerns with experimental and considerable editorial changes.

SUGGESTIONS TO AUTHORS

Main points:

1. The title claims Ubiquitin 2 regulates gag-pol degradation via E6AP, but most of the study was done with a triple knockout cell line that lacks three ubiquitin members. How can they conclude that it is Ubiquitin 2 that regulates gag-pol. The title also implies that E6AP (UBE3A) acts downstream of Ubiquitin 2, but there is no evidence for this.

It is true that most of the study is done with TKO cells. We concluded that it is Ubiquitin 2 that regulates gag-pol for a few reasons: (1) Single knockout of UBQLN1, UBQLN2, or UBQLN4 showed in Black and Hanson et al. (2023) that only UBQLN2 perturbation results in gag-pol accumulation, suggesting that UBQLN1 and UBQLN4 do not regulate PEG10. Similar data of UBQLN2 knockout from (Mohan et al., 2025) also suggests that deletion of UBQLN2 alone is sufficient to increase gag-pol. (2) Rescue of TKO cells with UBQLN2 from Black and Hanson et al. (2023) is sufficient to decrease gag-pol levels, and (3) co-IP data from this manuscript suggests that UBQLN2, but not UBQLN1, binds to gag-pol.

We agree that we have not shown directly that UBE3A- and ubiquitin-dependent proteasomal degradation of gag-pol is dependent on UBQLN2 binding to gag-pol first. Therefore we would be pleased to change the title to “UBQLN2 is necessary for UBE3A-mediated proteasomal degradation of the domesticated retroelement PEG10”.

2. Figure 1A should provide the drug concentration information.

We have added concentration information for Figure 1A in the figure legend.

The accumulation of gag-pol after 16h of proteasome inhibition is not very impressive. How do they know that this is due to protein stabilization as opposed to regulations at other levels (e.g. transcription, RNA stability, translation)?

It is true that there are many factors that could contribute to a steady-state level of a protein. We do not think that transcription or RNA stability contribute to UBQLN2-dependent accumulation of PEG10 because we see a specific accumulation of gag-pol, but not gag, which are both derived from the same mRNA.

We performed a protein synthesis assay in the presence or absence of proteasomal degradation and did not see meaningful differences in the PEG10 synthesis rate between WT and Ubiquitin knockout cells. If anything, we consistently saw that TKO cells made less gag-pol over 4 hours than WT cells, though this could not be easily quantified due to limitations in signal sensitivity in TKO

cells. This was repeated twice and is included as a new Figure S3.

The presentation of Fig 1B (and other quantification data) is not appropriate. The y-axis does not start from 0, which is misleading.

We agree that we do not want to mislead the reader and have altered all graphs as suggested.

3. Page 4, line 21, the authors claimed that only high molecular weight 120-250kDa signal was increased in cells treated with a proteasome inhibitor. However, from the data in Fig. 1C, it seems that in addition to the high molecular weight smear, there are two clear bands at ~90 and 100 kDa positions in bortezomib treated cells. There is no information on whether these bands are present in untreated cells.

Yes, we agree that Figure 1C shows those two bands at ~90 and 100 kDa very clearly. It was sometimes more difficult to visualize those two bands in particular, despite a higher molecular weight smear being present upon bortezomib treatment, but those two bands are never present in untreated cells. We have mentioned these bands on page 4 line 18 and on page 10 lines 22-24.

Fig. S1A and B quantified 4 different conditions, but the representative gel in C only showed 2 conditions.

It would be nice to include the gel data for the other two conditions.

We apologize for the confusion with Figure S1: S1A and S1B quantify data from Figure 1A. To shorten the manuscript and simplify our conclusions, we have removed S1A and S1B.

4. Figure 1C, was this IP experiment done under denaturing conditions? The experimental procedure does not include any information.

Experimental procedures for IPs were not done under denaturing conditions. We have edited our Methods section on page 14 line 30 to more clearly mention this.

5. When addressing the role of ubiquitin in gag-pol degradation (Fig. 2), they used an unconventional experimental design that paired two independent experiments together (Fig. 2D). A more straightforward way to look at this is to directly measure the half life of gag-pol in WT and Ubiquitin knockout cells using cycloheximide chase.

We regret the confusion with Figure 2. We did not pair two independent experiments together in Figure 2D - those data were all experimentally prepared and collected in pairs of WT and TKO samples and simply separated in Figures 1E and 2C for ease of interpretation. Historically, we have found that the audience is confused if we first present the data in Figure 2D. Instead, we have found that readers prefer to see the data one cell line at a time, and then summarize with the TKO/WT transformation of the data. This has been explained more thoroughly in the Figure 2 legend.

To directly address degradation rates, we performed a cycloheximide chase for 20 hours in WT and TKO cells for endogenous PEG10 with concomitant inhibition of the proteasome, autophagy, or ubiquitination. With three replicates, we found no differences between WT and TKO cells for gag-pol or gag degradation, as shown in a new Figure S2C-E.

We have reasons to believe that a cycloheximide chase may not be well suited for examination of UBQLN2 clients. First (1), PEG10 appears to have a long half-life, which makes cycloheximide chases difficult due to toxicity of cycloheximide at 24 hours. Second (2), PEG10 readily forms large oligomers and virus-like particles in cells (see Campodonico et al. 2024 and Mohan et al. 2025) that may be inaccessible to UBQLN2. Over time, following inhibition of protein synthesis, remaining PEG10 monomers may be more likely to coalesce, which would obscure the role of UBQLN2. Third (3), Ubiquilins have been implicated in quality control pathways (Juszkiewicz and Hegde, 2018; Muller et al., 2023). By shutting off translation with cycloheximide, we may have inhibited Ubiquilin-targeting to nascent PEG10 peptides. We have preliminary data that suggests that active translation of the *PEG10* mRNA plays a role in UBQLN2 targeting to PEG10. However, it is an incomplete story that we consider beyond the scope of this current manuscript, where

we would like to keep more focus on UBE3A.

6. Fig 2E shows that gag binds ubiquitin 2 much better than gag-pol. They should discuss why this substrate is not degraded.

We would agree that it appears that gag binds to UBQLN2 more than gag-pol does given the apparent difference in the gag:gag-pol ratio from cell lysate as compared to UBQLN2-coprecipitated PEG10. We have improved our discussion of why gag is not a client on page 10 lines 11-17.

7. Figure 3, they should explore how these lysine mutants affect the ubiquitination status of gag-pol. This is critical for the interpretation of their data because it is strange that mutating lysines in either the gag or pol region can "stabilize" this protein. In general, if lysines play redundant roles in protein turnover, one has to mutate all of them to see a phenotype. It is possible that some lysine mutations may affect protein via a ubiquitin independent mechanism.

We agree that this is an important question to address.

In Figure 4D, we looked at the ubiquitination status of two mutant proteins, gag^{K>R}-pol and gag-pol^{K>R} upon co-immunoprecipitation of UBQLN2. In that blot, we found that gag-pol^{K>R} seemed to show less ubiquitination than gag^{K>R}-pol, suggesting that most of the visible ubiquitination events occur in the pol lysines. This was not previously addressed, so we have added text on pages 7 lines 30-35 and page 10 lines 26-28 to point out this phenomenon.

We examined the possibility that two other common lysine modifications, acetylation and ADP ribosylation, might occur on gag-pol protein. To test this, we performed western blot of immunoprecipitated PEG10 protein and probed for the two modifications with well-described antibodies. While both antibodies showed positivity in cell lysate, we did not see any marking on PEG10, which led us to the conclusion that it is unlikely that PEG10 is modified by acetylation or ADP ribosylation in a significant way. This, paired with our TAK243 and ubiquitin blot data, leads us to conclude that the high molecular weight PEG10 is very likely ubiquitinated protein. This is summarized in Supplemental Figure 1A-B and discussed on page 4 lines 21-25.

8. Page 8, lines 15-38, the lengthy introduction about UBE3A should be placed in the introduction section.

We agree it flows much better this way, and thank you for the suggestion. The introduction of UBE3A now occurs on page 3 lines 4-13 and has been shortened.

9. Please provide representative blots for Fig. 5E.

Figure 6 has been updated to show higher molecular weight products, although the discussion of high molecular weight PEG10 has been removed for clarity and length.

Does UBE3A knockdown affect the low molecular weight ubiquitinated gag-pol species?

We regret that this was not clear. Figure 6D is looking at the 80 kDa form of gag-pol, some of which we think is ubiquitinated, as visualized in Figure 1C.

Why did knockdown of an ubiquitin ligase cause substrate to accumulate in ubiquitinated form. This result raises the question of the role of UBE3A in gag-pol regulation and suggests that the model in their graphic abstract is oversimplified.

We apologize that our interpretation of this counterintuitive result was not clear. Because the absence of UBE3A resulted in a paradoxically higher proportion of high molecular weight gag-pol, we concluded that UBE3A promotes a form of ubiquitination that results in proteasomal degradation, but that there may be additional E3 ligases that promote ubiquitination of gag-pol

that does not result in proteasomal degradation.

However, because of the risk of over-interpretation of the data, and the additional work we'd like to do to investigate this thoroughly, we have removed explicit discussion of high molecular weight gag-pol from our manuscript.

10. The co-IP experiment does not prove a direct interaction between UBE3A and gag-pol. There is no evidence to suggest that Ubiquilin, UBE3A and substrate can form a ternary complex.

We agree that we have not yet shown conclusive evidence that Ubiquilin, UBE3A, and substrate form a ternary complex. We have been more precise about how UBE3A co-precipitates with UBQLN2 when referencing our data.

11. The paper should be shortened and revised to improve the clarity and readability.

We appreciate this comment and have revised and shortened the manuscript as suggested.

The authors should pay close attention to their conclusions to ensure they are logical and without over-interpretation. For example, on page 11, lines 21-23. These sentences are confusing. I don't understand what the authors meant by "there were comparatively minimal changes to TKO cells". Didn't they just show that TKO cells had more gag-pol in Fig. 5A? How can they conclude here that UBE3A functions downstream of UBQLN2? The next sentence is even more confusing. They claimed that "UBQLN2 is necessary for UBE3A-mediated ubiquitination of PEG10". Where is the evidence? The data on Fig. 5E clearly showed the opposite (the knockdown of UBE3A INCREASED gag-pol ubiquitination). The data is against their model in the graphic abstract.

We regret that our conclusions may have come across as over-interpreted in some places and have attempted to simplify and streamline our discussion in response.

The discussion on the liquid-liquid phase separation of ubiquilin is completely irrelevant to the study.

We have removed this discussion from the manuscript.

Reviewer 2: SUMMARY OF THE ADVANCE MADE IN THIS PAPER AND ITS POTENTIAL SIGNIFICANCE TO THE FIELD

Ubiquilins are a family of UBL/UBA containing proteins that are known to shuttle ubiquitinated proteins to the proteasome for degradation. Building off previous work showing that the domesticated retrotransposon PEG10 is a UBQLN2 client, Roberts et al provide a detailed analysis of the different steps for UBQLN2 mediated degradation of PEG10. Key results include demonstrating that Ubiquilins mediate proteasomal degradation of PEG10, showing that PEG10-UBQLN2 interaction is ubiquitin-independent, identification of key lysine residues in PEG10, and the discovery that the UBE3A-UBQLN2 interaction facilitates PEG10 ubiquitination. Overall, I find the results to be relatively rigorous and, after revision, appropriate for publication. Given the complexity of the system and the mixed results (such as proteasomal vs. autophagy pathways, multiple E3 ligases, multiple ubiquitination events required), the paper could benefit from rewriting to better acknowledge nuance in the system, and some clarifying experiments.

Thank you to this reviewer for their feedback and helpful suggestions, which we have addressed below.

SUGGESTIONS TO AUTHORS

Major Concerns

* While it is clear that some portion of gag-pol is degraded via the proteasome and Ubiquilins mediate this function, I am not convinced that the proteasome is the major degradation pathway.

Instead, the results in figure 1E indicate that the majority of gag-pol degradation may be occurring via autophagy. Many substrates are known to show plasticity in which degradation pathway is utilized, with a shift occurring upon inhibition of one pathway. The following changes are recommended.

- o Repeat the experiments in 1A-C with chloroquine.

We thank the reviewer for pushing us to include an experiment examining the contribution of autophagy to endogenous PEG10 degradation. We repeated the experiments from Figure 1 with chloroquine and these are now included as Figure 2C-D and Figure S2A. Interestingly, we found that endogenous PEG10 protein was not significantly regulated by autophagy for either WT or TKO cells, in contrast with our overexpressed PEG10 assay in Figure 1D-E. We hypothesize that autophagy may be more important in regulating the degradation of transfected PEG10 protein as opposed to endogenous levels, but this warrants further work.

- o Identify the ubiquitin linkage on PEG10 after proteasomal inhibition. Is it K48 linked? This is optional, but it could provide some clarity about proteasomal degradation vs. autophagy.

Western blots have been mixed on this but suggest the majority is K48. We have not included it in the manuscript because we found that the low signal was impossible to quantify and gave rather variable results, but have included a draft figure here. We have concluded that we'll need to perform ubiquitin-linked proteomics to get the best answer about the location and nature of ubiquitin on PEG10 proteins.

"[NOTE: an image provided in confidence for the reviewers has been removed]"

- o Language will need to change to indicate that while degradation can occur via the UPS, this is not the exclusive (or even dominant) pathway.

We agree that this nuance was overlooked in our discussion and thank the reviewer for pointing this out. We have changed language in the Figure legends of Figure 1 and Figure 2, as well as text in the Results to reflect this.

* An alternative model is that the gag to gag-pol ratio is changing due to differences in translation rather than differences in protein stability. It is difficult to differentiate between these models because figure 1 only looks at steady state levels. The following changes are recommended.

- o Given the extensive comparisons between the stability of gag vs. gag-pol, the article would benefit from enhanced introduction of translation of PEG10, including what determines the ratio of translation gag vs. gag-pol protein.

We have included more information about the -1 PRF of PEG10 on page 3 lines 21-27 and explored synthesis of gag and gag-pol directly using metabolic labeling in WT and TKO cells, which is summarized in Figure S3.

- o Unless the gag to gag-pol translation ratio is known to be stable across many stress conditions, Figure 1 should be repeated as a pulse-chase or cycloheximide chase to look at protein stability.

We performed a cycloheximide chase which is included as Figure S2C-E. It did not find a difference in the degradation rate between WT and TKO cells; however, there are multiple reasons why we think a chase like this is unlikely to identify differences in PEG10 breakdown, as mentioned above.

- o I know that the authors have used the Dendra/CFP assay previously to study PEG10 stability. However, the figure and description could benefit from some clarification. For example, it's not readily clear how gag levels were measured in figure 1E. I assume it is a separate construct that only encodes Gag-Dendra and lacks the Pol region.

Yes, indeed; we have a construct that encodes gag-Dendra but lacks the pol region. We

apologize that this was not clear and have updated Figure 1 and text.

Minor Concerns

* Please label western blots to indicate which antibody is used (HA, Flag, gag-pol, UBQLN2, etc.). I know this is in the figure legend, but it would help if it is directly labeled in the figure.

This has been updated for ease of interpretation; thank you for the suggestion.

* Statistics for many of the bar graphs are far away from the bar and it takes some effort to match the statistics with the appropriate bars (for example, Figure 2B). If possible, please adjust.

This has also been updated for ease of interpretation.

* Figure legend 1A states that this is a representative experiment twice.

This has been fixed in the figure legend.

* In figure 1B, are the results with gag statistically significant? Perhaps explicitly state that only significant values are labeled.

They were not statistically significant and we have updated the Figure legend to more clearly state this.

* Please add a supplemental figure showing representative flow cytometry data for figure 1E and 2C. It is important to see the Dendra2/CFP ratio for the entire population of cells rather than just the average.

We have generated new supplemental figures showing representative flow cytometry data for Figure S1C and Figure S2B.

* Please be more explicit for how 2D was prepared. For example, it appears that it is just the Dendra/CFP ratio for TKO/WT. Or is it the (Dendra/CFP)/untreated for TKO/WT?

It was the former: Dendra/CFP ratio for TKO/WT. This has been clarified in the Figure legend.

* Figure 6A, please indicate number of replicates.

Thank you for noticing; this has been updated in the Figure legend of what is now Figure 5.

* The abstract states that ubiquitination is dependent of Ube3a, but the results show that other E3 ligases are involved. The abstract should be rewritten accordingly.

We have updated the abstract to reflect a more general influence of UBE3A on the UPS-dependent degradation of gag-pol protein.

* The authors frequently state that gag is not a ubiquilin client, but the data clearly shows gag interacting with UBQLN2. The authors should be explicit about what it means to be a client (shuttled for degradation vs. binding).

We have updated the results on page 7 lines 10-13, and in the discussion on page 10 lines 11-17 to better discuss this interesting phenomenon.

Reviewer 3: SUMMARY OF THE ADVANCE MADE IN THIS PAPER AND ITS POTENTIAL SIGNIFICANCE TO THE FIELD

This study investigates the degradation of the gag-pol fusion protein encoded by PEG10. Earlier work had shown that PEG10 degradation occurs via UBQLN2 and the proteasome. In the current work, the authors build on this in order to gain additional insights into how UBQLN2 selects clients and how it targets them for degradation. The field is a bit muddled on this issue, with models

ranging from UBQLNs selecting clients after they have been ubiquitinated, to ones in which UBQLNs select clients and facilitate their ubiquitination. The most recent, and arguably most rigorous and compelling, studies indicate that UBQLNs using their STI domains to recognise exposed hydrophobicity, then use the UBA domain to recruit E3 ligase(s), and finally use the UBL domain to target the substrate to the proteasome. This model was elucidated in PMID 27345149, with further structural analysis of the STI domains in PMID 33620121 & 39026758, and characterisation of the client range in PMID 28933694.

In the context of current knowledge, the main advance of this work is to characterise another substrate (PEG10) and implicate UBE3A as one of the Ubiquilin-dependent ligases involved in substrate degradation. The characterisation of PEG10 degradation as being UBQLN, ubiquitin, and proteasome dependent is generally clear, but is a modest advance beyond earlier work. The latter part on UBE3A is a bit underdeveloped. If this can be improved, and a few additional controls provided as noted below, the paper should be suitable for publication.

We thank the reviewer for these thoughtful comments and the opportunity to improve our manuscript's strength. We have performed the new experiments suggested by the reviewer, including generation of a new CRISPR cell line, to address their recommendations.

SUGGESTIONS TO AUTHORS

Specific points:

1) (minor) Fig. 1 and 2 are baseline characterisation of PEG10 degradation, and are generally clear and convincing. However, it would be useful to have had direct comparisons of WT versus TKO cells analysed on the same gel in at least some of these initial experiments so a reader can directly appreciate the magnitude of the difference in steady state levels relative to suitable loading controls. This can be seen in Fig. 5A, but as a reader, I would like to have seen such a direct comparison early in the paper.

We agree that the best comparison is with WT and TKO samples on the same gel. We have included a new Figure 2C-D and Figure S2A, with the added benefit of also including chloroquine as a drug treatment for endogenous PEG10 levels.

2) (major) Figure 4 would benefit from analysis of a UBQLN2 variant in which one or more of the STI domains are deleted to directly test whether these regions are needed as hypothesized. This is an important conclusion of the study - that client recognition is not via ubiquitin or the UBA domain - so actually identifying the region of UBQLN2 that recognises clients seems important and central to the study.

We completely agree with this and thank the reviewer for bringing this up, as it provided us with unexpected findings that we think strengthen the manuscript. Recent work shows quite convincingly that the STI1 domain(s) of Dsk2 and Ubiquilins play a major role in binding to select client proteins (Acharya et al., 2025; Liu et al., 2025; Onwunma et al., 2024; Suzuki and Kawahara, 2016) and we would hypothesize the same for PEG10 and UBQLN2. We therefore deleted both STI1 domains of UBQLN2 and performed crosslinking co-immunoprecipitations for PEG10.

We were surprised to find that deletion of the STI1 domains was dispensable for PEG10 binding, which is now included in Figure 4E. This would suggest that PEG10 binding to UBQLN2 is distinct from recently described reports of client:Ubiquilin interactions being dominated by STI1 domains of the Ubiquilin.

With this in mind, we went a step further to characterize the contribution of the PXX domain to the UBQLN2:PEG10 interaction. Here, we found a more dramatic loss of PEG10 binding when the PXX domain was deleted, which is in Figure 4F.

We think this may reflect the unique relationship between UBQLN2 and PEG10. Our group and others have found that UBQLN2, but not UBQLN1, uniquely regulates PEG10 abundance. In this study, we also find that UBQLN1 does not bind to PEG10. The STI1 domains of UBQLN1 and

UBQLN2 are remarkably similar; as such, it would be somewhat surprising if PEG10 bound to UBQLN2 via ST11 domain interactions, but was incapable of binding to UBQLN1. We hypothesize that the unexpected reliance of the UBQLN2:PEG10 interaction on the PXX domain, and not the ST11 domains, may explain why PEG10 has such a specific relationship with UBQLN2 and not other Ubiquilins. This is summarized on page 8 lines 2-12 and on pages 10-11 lines 36-10.

3) (major) The siRNA result for UBE3A is not entirely convincing without a rescue. It would be better if UBE3A were knocked out with CRISPR and rescued with either wild type or Δ AZUL versions of UBE3A. This will not only establish specificity of the effect, but would also test whether the AZUL domain is functionally important. As the UBE3A part of the study is the main new advance, it seems reasonable to establish this point more thoroughly and to provide at least some basic molecular insights into which part of UBE3A interacts with which part of UBQLN2.

We agree that a rescue would show more convincingly that UBE3A regulates the abundance of PEG10, with the opportunity to examine dependence on the AZUL domain. We generated two new CRISPR cell lines of TKO HEK293 cells lacking UBE3A and rescued them with either WT or delta AZUL UBE3A in the presence or absence of co-transfected UBQLN2.

These findings are summarized in new Figure 6E-G and Figure S6E-F. They are discussed on page 9 lines 12-30.

4) (major) Related to point 3, the prediction is that Ubiquilin's UBA domain is responsible for interacting with UBE3A to facilitate substrate ubiquitination. This seems worthwhile to test by analysing the interaction of WT versus Δ UBA versions of UBQLN2 in co-IP experiments.

Indeed, we thought that the UBA domain would be responsible for binding to UBE3A, and we thank the reviewer for this push to test the hypothesis directly. We tested this in Figure 5C and found that deletion of the UBA domain did not ablate UBE3A binding. We think that this may be due to contribution of UBA-adjacent sequences to UBE3A binding, which is alluded to in (Buel et al., 2023) and is mentioned on pages 8-9 lines 35-2.

References

- Acharya, N., Daniel, E. A., Dao, T. P., Niblo, J. K., Mulvey, E., Sukenik, S., Kraut, D. A., Roelofs, J. and Castañeda, C. A. (2025). ST11 domain dynamically engages transient helices in disordered regions to drive self-association and phase separation of yeast ubiquilin Dsk2.
- Buel, G. R., Chen, X., Myint, W., Kayode, O., Folimonova, V., Cruz, A., Skorupka, K. A., Matsuo, H. and Walters, K. J. (2023). E6AP AZUL interaction with UBQLN1/2 in cells, condensates, and an AlphaFold-NMR integrated structure. *Structure* **31**, 395-410.e6.
- Juszkiewicz, S. and Hegde, R. S. (2018). Quality Control of Orphaned Proteins. *Molecular Cell* **71**, 443-457.
- Liu, D., Huo, X.-Y., Zhang, X. and Zhang, Z.-R. (2025). The E3 ubiquitin ligase RNF126 facilitates quality control of unimported mitochondrial membrane proteins. *Journal of Biological Chemistry* **301**, 108403.
- Mohan, H. M., Fernandez, M. G., Huang, C., Lin, R., Ryou, J. H., Seyfried, D., Grotewold, N., Barget, A. J., Whiteley, A. M., Basrur, V., et al. (2025). Endogenous retrovirus-like proteins recruit UBQLN2 to stress granules and shape their functional biology. *Sci. Adv.* **11**, eadu6354.
- Muller, M. B. D., Kasturi, P., Jayaraj, G. G. and Hartl, F. U. (2023). Mechanisms of readthrough mitigation reveal principles of GCN1-mediated translational quality control. *Cell* **186**, 3227-3244 e20.

Onwunma, J., Binsabaan, S., Allen, S. P., Sankaran, B. and Wohlever, M. L. (2024). ALS mutations disrupt self-association between the Ubiquilin Sti1 hydrophobic groove and internal placeholder sequences.

Suzuki, R. and Kawahara, H. (2016). UBQLN4 recognizes mislocalized transmembrane domain proteins and targets these to proteasomal degradation. *EMBO Rep* 17, 842-857.

Second decision letter

MS ID#: jcs.264105R1

MS TITLE: UBQLN2 is necessary for UBE3A-mediated proteasomal degradation of the domesticated retroelement PEG10

AUTHORS: Julia Roberts; Phuoc Huynh; Luis Carale; Alexandra Whiteley
ARTICLE TYPE: Research Article

Dear Dr Whiteley,

We have now reached a decision on the above manuscript.

As you will see, the reviewers gave favourable reports but raised some critical points that will require amendments to your manuscript. I hope that you will be able to carry these out because I would like to be able to accept your paper, depending on further comments from reviewers. Specifically, it would be important to address the remaining concern of reviewer #1, either by modifying the text or performing the suggested experiment.

SUMMARY OF THE ADVANCE MADE IN THIS PAPER AND ITS POTENTIAL SIGNIFICANCE TO THE FIELD

Ubiquilins are a family of UBL/UBA containing proteins that are known to shuttle ubiquitinated proteins to the proteasome for degradation. Building off previous work showing that the domesticated retrotransposon PEG10 is a UBQLN2 client, Roberts et al provide a detailed analysis of the different steps for UBQLN2 mediated degradation of PEG10. Key results include demonstrating that Ubiquilins mediate proteasomal degradation of PEG10, showing that

PEG10-UBQLN2 interaction is ubiquitin-independent, identification of key lysine residues in PEG10, and the discovery that the UBE3A-UBQLN2 interaction facilitates PEG10 ubiquitination. Overall, I find the results to be relatively rigorous and, after revision, appropriate for publication. Given the complexity of the system and the mixed results (such as proteasomal vs. autophagy pathways, multiple E3 ligases, multiple ubiquitination events required), the paper could benefit from rewriting to better acknowledge nuance in the system, and some clarifying experiments.

SUGGESTIONS TO AUTHORS

Major Concerns

* While it is clear that some portion of gag-pol is degraded via the proteasome and Ubiquilins mediate this function, I am not convinced that the proteasome is the major degradation pathway. Instead, the results in figure 1E indicate that the majority of gag-pol degradation may be occurring via autophagy. Many substrates are known to show plasticity in which degradation pathway is utilized, with a shift occurring upon inhibition of one pathway. The following changes are recommended.

o Repeat the experiments in 1A-C with chloroquine.

- o Identify the ubiquitin linkage on PEG10 after proteasomal inhibition. Is it K48 linked? This is optional, but it could provide some clarity about proteasomal degradation vs. autophagy.
- o Language will need to change to indicate that while degradation can occur via the UPS, this is not the exclusive (or even dominant) pathway.
- * An alternative model is that the gag to gag-pol ratio is changing due to differences in translation rather than differences in protein stability. It is difficult to differentiate between these models because figure 1 only looks at steady state levels. The following changes are recommended.
 - o Given the extensive comparisons between the stability of gag vs. gag-pol, the article would benefit from enhanced introduction of translation of PEG10, including what determines the ratio of translation gag vs. gag-pol protein.
 - o Unless the gag to gag-pol translation ratio is known to be stable across many stress conditions, Figure 1 should be repeated as a pulse-chase or cycloheximide chase to look at protein stability.
 - o I know that the authors have used the Dendra/CFP assay previously to study PEG10 stability. However, the figure and description could benefit from some clarification. For example, it's not readily clear how gag levels were measured in figure 1E. I assume it is a separate construct that only encodes Gag-Dendra and lacks the Pol region.

Minor Concerns

- * Please label western blots to indicate which antibody is used (HA, Flag, gag-pol, UBQLN2, etc.). I know this is in the figure legend, but it would help if it is directly labeled in the figure.
- * Statistics for many of the bar graphs are far away from the bar and it takes some effort to match the statistics with the appropriate bars (for example, Figure 2B). If possible, please adjust.
- * Figure legend 1A states that this is a representative experiment twice.
- * In figure 1B, are the results with gag statistically significant? Perhaps explicitly state that only significant values are labeled.
- * Please add a supplemental figure showing representative flow cytometry data for figure 1E and 2C. It is important to see the Dendra2/CFP ratio for the entire population of cells rather than just the average.
- * Please be more explicit for how 2D was prepared. For example, it appears that it is just the Dendra/CFP ratio for TKO/WT. Or is it the (Dendra/CFP)/untreated for TKO/WT?
- * Figure 6A, please indicate number of replicates.
- * The abstract states that ubiquitination is dependent of Ube3a, but the results show that other E3 ligases are involved. The abstract should be rewritten accordingly.
- * The authors frequently state that gag is not a ubiquilin client, but the data clearly shows gag interacting with UBQLN2. The authors should be explicit about what it means to be a client (shuttled for degradation vs. binding).

Revision 1:

SUMMARY OF THE ADVANCE MADE IN THIS PAPER AND ITS POTENTIAL SIGNIFICANCE TO THE FIELD

The authors have thoughtfully addressed all concerns I raised during the initial review. I recommend acceptance for publication.

SUGGESTIONS TO AUTHORS

Major comments [Please request additional experiments only if they are essential for supporting the conclusions; authors should be encouraged to highlight any claims that are preliminary or speculative, or to discuss any pitfalls or alternative interpretations in a 'Limitations' section]

Minor comments

Second revision

Author response to reviewers' comments

We thank the reviewers for their constructive comments and thorough analysis of our manuscript in this second round of review, and have made additional changes below.

Also, we have reformatted our Funding Statement and have now included the Competing Interests section in our manuscript text. We also moved the large blot transparency file and appended it to the Supplementary Material.

Comments from the Reviewers:

Reviewer 1: SUMMARY OF THE ADVANCE MADE IN THIS PAPER AND ITS POTENTIAL SIGNIFICANCE TO THE FIELD

In the revised manuscript, the authors have added new data and modified the text, which satisfactorily addresses most of my previous concerns. However, several important issues remain unresolved.

SUGGESTIONS TO AUTHORS

The authors acknowledge that the ubiquitination assay shown in Figure 1C was performed under native conditions. Consequently, the assay cannot discriminate between direct ubiquitination of Gag-Pol and ubiquitination of a Gag-Pol-associated protein. The same concern applies to Figure 4D, where certain Gag-Pol lysine mutants co-precipitate with reduced levels of ubiquitin conjugates. These conjugates could be on interacting proteins rather than on Gag-Pol itself; the lysine mutations could alter the binding of Gag-Pol to these partners, indirectly reducing ubiquitin signals. This technical ambiguity undermines the authors' main conclusion regarding direct ubiquitination of Gag-Pol. The issue must be resolved experimentally—such as by performing denaturing immunoprecipitation or other assays that can directly detect ubiquitinated Gag-Pol—to substantiate the central claim.

We thank the reviewer for pointing out this limitation of the data presented in Figures 1C and 4D. We agree that the use of Triton-X buffer for our lysis means that it is possible that the ubiquitin signal present upon HA-tag immunoprecipitation could be from a co-precipitating protein. We considered this unlikely because of the strong concordance of HA-tag signal and ubiquitin signal laddering, which we have tried to highlight in the text of the Results. For Figure 4D, we made a calculated decision to perform a non-denaturing co-precipitation because of the added benefit of assessing UBQLN2 binding upon K>R mutation of PEG10.

It is also possible that K>R mutations induce unexpected structural changes, though we think that this is an unlikely contributor. K>R mutation did not influence UBQLN2 co-precipitation, which we think highlights how unlikely it is that the K>R mutations induce structural changes that alter binding to undiscovered partners.

We agree that it is important to acknowledge these concerns. We have altered the text in the results section in multiple places to mention these limitations, and have added multiple sentences in the Discussion where we further discuss them.

Reviewer 2: SUMMARY OF THE ADVANCE MADE IN THIS PAPER AND ITS POTENTIAL SIGNIFICANCE TO THE FIELD

The authors have thoughtfully addressed all concerns I raised during the initial review. I recommend acceptance for publication.

SUGGESTIONS TO AUTHORS

Major comments

Thank you to this reviewer for their assessment of our work.

Reviewer 3: The authors have addressed my comments in full. I appreciate their attention to these items, and feel the paper is now substantially stronger.

Thank you to this reviewer for their assessment, and for the opportunity to strengthen our paper with their recommendations, which we agree improved the paper substantially.

Third decision letter

MS ID#: jcs.264105R2

MS Title: UBQLN2 is necessary for UBE3A-mediated proteasomal degradation of the domesticated retroelement PEG10

Authors: Julia Roberts; Phuoc Huynh; Luis Carale; Alexandra Whiteley
Article Type: Research Article

Dear Dr Whiteley,

I am happy to tell you that your manuscript has been accepted for publication in Journal of Cell Science, pending standard publication integrity checks.